# Identification of an innexin required for termination of the asexual state in planarian reproductive switching

Nobuyoshi Kumagai[1] Michio Kuroda[1], Tosei Hanai[1], Masaki Fujita[1], Takaaki Hino[1], Shunta Yorimoto[2], Sayaka Manta[1], Shuzo Nakagawa[1], Moe Yokoyama[1], Leon Tajima[1], Riku Ito[1], Hikaru Yamada[1], Kota Miura[1], Makoto Kashima[3], Katsushi Yamaguchi[4], Shuji Shigenobu[2,4], Ryohei Furukawa[5], Kiyono Sekii[6]*, Kazuya Kobayashi[1]*

1 Department of Biology, Faculty of Agriculture and Life Science, Hirosaki University, Hirosaki, Aomori Japan, 2 Life Science Center for Survival Dynamics, Tsukuba Advanced Research Alliance (TARA), University of Tsukuba, Tsukuba, Japan, 3 Department of Biomolecular Science, Faculty of Science, Toho University, Funabashi, Chiba, Japan, 4 NIBB Core Facility, National Institute for Basic Biology, Okazaki Japan, 5 Department of Biology, Research and Education Center for Natural Sciences, Keio University, Yokohama, Kanagawa, Japan, 6 Faculty of Business and Commerce, Keio University, Yokohama, Kanagawa, Japan

* kobkyram@hirosaki-u.ac.jp (KK); kiyono.sekii@gmail.com (KS)

## Abstract

Many metazoans switch between asexual and sexual reproduction based on environmental changes, life cycle phases, or both. This reproductive strategy enables them to benefit from the features of both reproductive modes. In general, asexual reproduction is broadly divided into parthenogenesis and vegetative reproduction. As in parthenogenesis, individuals develop ovaries and lay eggs, the most significant event in switching from parthenogenesis to sexual reproduction is the production of testes. Meanwhile, in vegetative reproduction, individuals do not need germ cells themselves. Thus, they must post-embryonically develop and maintain germ cells derived from pluripotent cells as they switch from vegetative to sexual reproduction. The complicated mechanisms for controlling the postembryonic reproductive development remain unknown. The planarian *Dugesia ryukyuensis* can switch from vegetative to sexual reproduction by stimulating bioactive compounds called sex-inducing substances, which are widely conserved in Platyhelminthes, including parasitic flatworms. The two reproductive modes are facilitated by the presence of adult pluripotent stem cells, which generate any type of somatic tissue in the asexual state and produce and maintain hermaphroditic reproductive organs in the sexual state. In this study, using RNA sequencing analysis in experimental sexualization by sex-inducing substances, we identified four essential genes for sexualization. A common feature following the knockdown of the four essential genes was a blockage of testicular differentiation. One of the four essential genes was a gap junction gene, *Dr-siri* (*Dugesia ryukyuensis*-<u>s</u>exual <u>i</u>nduction-<u>r</u>elated <u>i</u>nnexin). We suggest that the

**Data availability statement:** RNA sequencing raw data generated in this study have been deposited in the DNA Data Bank of Japan Sequence Read Archive (DRA; https://www.ddbj.nig.ac.jp/dra/index-e.html) under accession numbers DRR747365–DRR747382 for the libraries derived from worms during sexualization, and DRR707054–DRR707103 for the libraries derived from Dr-nhr-1 (RNAi), Dr-dmd-1 (RNAi), and Dr-klf4l (RNAi) worms.

**Funding:** This work was supported by Hirosaki University Institutional Research Grant for Future Innovation and a Grant-in-Aid for Scientific Research (KAKENHI; grant numbers 16H01249 [KK], 19H03256 [KK], 19H05236 [KK] and 25K02303 [KK]) from the Ministry of Science, Culture, Sports, and Education, Japan. The funders had no role in study design, data collection and analysis, decision to publish, or preparation of the manuscript.

**Competing interests:** The authors have declared that no competing interests exist.

establishment of a testicular stem cell niche supported by Dr-siri protein is responsible for the breakthrough of dormancy in postembryonic reproductive development in planarian reproductive switching. Our findings suggest that the production of testes might be crucial for even switching from vegetative to sexual reproduction.

## Author summary

Some animals have pluripotent stem cells that can differentiate into all types of cells that make up their adult bodies. Thus, they can regenerate after self-amputation and reproduce asexually to increase the number of individuals identical to themselves. Additionally, they can reproduce sexually by producing eggs and sperm (gametes) from pluripotent stem cells at any time. In nature, they achieve procreation through a reproductive strategy that alternates between asexual reproduction, which allows for population growth, and sexual reproduction, which leads to various offspring. In this study, we identified four genes that are essential for the transition from asexual to sexual reproduction in the hermaphroditic species planaria. Three genes are predicted to function as transcription factors. Another gene was identified as a novel innexin—a gap junction protein that forms a small channel directly connecting somatic and germ cells, playing a crucial role in cell-to-cell communication. These four genes are implicated in testis differentiation, suggesting that the production of testes might play an important role in the switch from asexual to sexual reproduction in planarians.

## Introduction

William (1966) [1] defined sexual reproduction as the production of offspring "with new combinations of the parental genes". These new gene combinations typically result from the meiotic production of haploid gametes and their subsequent syngamy with gametes from two different individuals. According to this definition, various parthenogenetic reproduction modes (e.g., automixis, endomitosis, and apomixis) can be considered as asexual modes. Meanwhile, various metazoans in which adult pluripotent stem cells (aPSCs) or the differentiated tissues that give rise to pluripotent cells are present can undergo vegetative reproduction by budding, fission, or fragmentation, accompanied by regeneration [2–7]. Vegetative reproduction is asexual reproduction in a strict sense, namely amixis (reproduction without meiosis and syngamy). This reproduction is widespread in many animal phyla, including Porifera, Placozoa, Cnidaria, Platyhelminthes, Entoprocta, Sipuncula, Annelida, Phoronida, Ectoprocta, Echinodermata, Hemichordata, and Chordata [8,9].

Mammals do not seem to have adult pluripotent stem cells and do not spontaneously undergo parthenogenesis due to genomic imprinting mechanisms [10,11]. In contrast, bdelloid rotifers have not shown any evidence of sexuality for tens of millions of years [12–14]. However, such organisms performing only either obligate

sexual or obligate asexual reproduction may be exceptional. Many metazoans occasionally switch between sexual reproduction and parthenogenesis or between sexual and vegetative reproduction based on environmental changes, life cycle phases, or both. Switching of the reproductive modes appears to be beneficial to their reproductive success by taking advantage of both sexual and asexual reproduction.

Mechanisms that control switching between sexual reproduction and parthenogenesis, specifically cyclic parthenogenesis, are extensively studied in cladoceran crustaceans using experimental systems with chemical stimuli such as juvenile hormone analogs [15–19]. In the water flea *Daphnia*, the appearance of males allows them to switch sexual reproduction from parthenogenesis, and a doublesex gene, which was a transcription factor containing an evolutionarily conserved domain, the Dsx/Mab-3(DM) domain, was identified as a gene responsible for the production of males [20].

Meanwhile, in vegetative reproduction, individuals must post-embryonically develop and keep germ cells from aPSCs or the differentiated tissues that give rise to pluripotent cells, when they switch from vegetative to sexual reproduction. Switching between sexual reproduction and vegetative reproduction via environmental stimulation is likely unstable in laboratory studies and is not feasible as a relatively quick and reliable assay system to study the underlying mechanisms. Therefore, the complicated mechanisms for controlling postembryonic reproductive development in switching from vegetative to sexual reproduction remain poorly understood.

Freshwater planarians (Platyhelminthes, Turbellaria, Seriata, Tricladida) have aPSCs that proliferate in the mesenchymal space and can differentiate into any cell type [21–28]. They exhibit a remarkable capacity for regeneration owing to planarian aPSCs. Furthermore, because of this strong regenerative capacity, some of them can undergo vegetative reproduction, which is achieved by successive biological processes from transverse fission to regeneration [29–32]. Planarians can also post-embryonically produce hermaphroditic germ cells and genital organs from aPSCs. These characteristics allow some planarians to switch between an asexual and a sexual state in response to environmental changes, particularly temperature changes [33–36]. Asexual worms develop reproductive organs even without being exposed to environmental stimuli if they are fed minced sexually mature worms of the same or different planarian species, suggesting that a sex-inducing substance(s) is present in sexually mature worms [37–43]. Through recent studies in *Dugesia ryukyuensis*, we found that sex-inducing substances are also widespread in parasitic flatworms [44]. We hypothesized that unidentified sex-inducing substances could be crucial for understanding planarian sexualization, similar to how juvenile hormone analogs have helped identify key genes involved in the shift from parthenogenesis to sexual reproduction.

In our assay system for sexual induction, asexual *Dugesia ryukyuensis* worms of the OH strain develop hermaphroditic reproductive organs in a regular sequence in approximately 1 month, when they are daily fed with food containing sex-inducing substances. The morphological changes during sexual induction allowed us to divide the process into six stages (Fig 1A) [45]. Sexual induction has a "point of no return" between stages 2 and 3. The worms in stages 1 and 2 return to being asexual if the administration of the sex-inducing substances is stopped, whereas from stage 3 onward, the worms keep developing reproductive organs, even if the administration of sex-inducing substances is halted [41]. Before reaching the critical point of no return, asexual OH worms, despite having pluripotent stem cells, do not develop reproductive organs beyond immature ovaries, indicating that genes functioning at this critical juncture are crucial for breaking dormancy in postembryonic reproductive development, although the molecular mechanisms remain unclear. This study aims to identify the essential genes involved in this process by constructing an RNA-seq library focused on the point of no return. Herein, we identified four essential genes for sexualization, three of which are transcriptional factor genes, and one is a gap junction gene. We propose an asexual state termination model involving these essential genes.

## Results

### RNA-seq analysis focusing on the point of no return

To identify the essential genes required for transgressing the point of no return, we prepared RNA-seq libraries focusing on the point of no return. The development of the external copulatory apparatus is one of the obvious criteria for

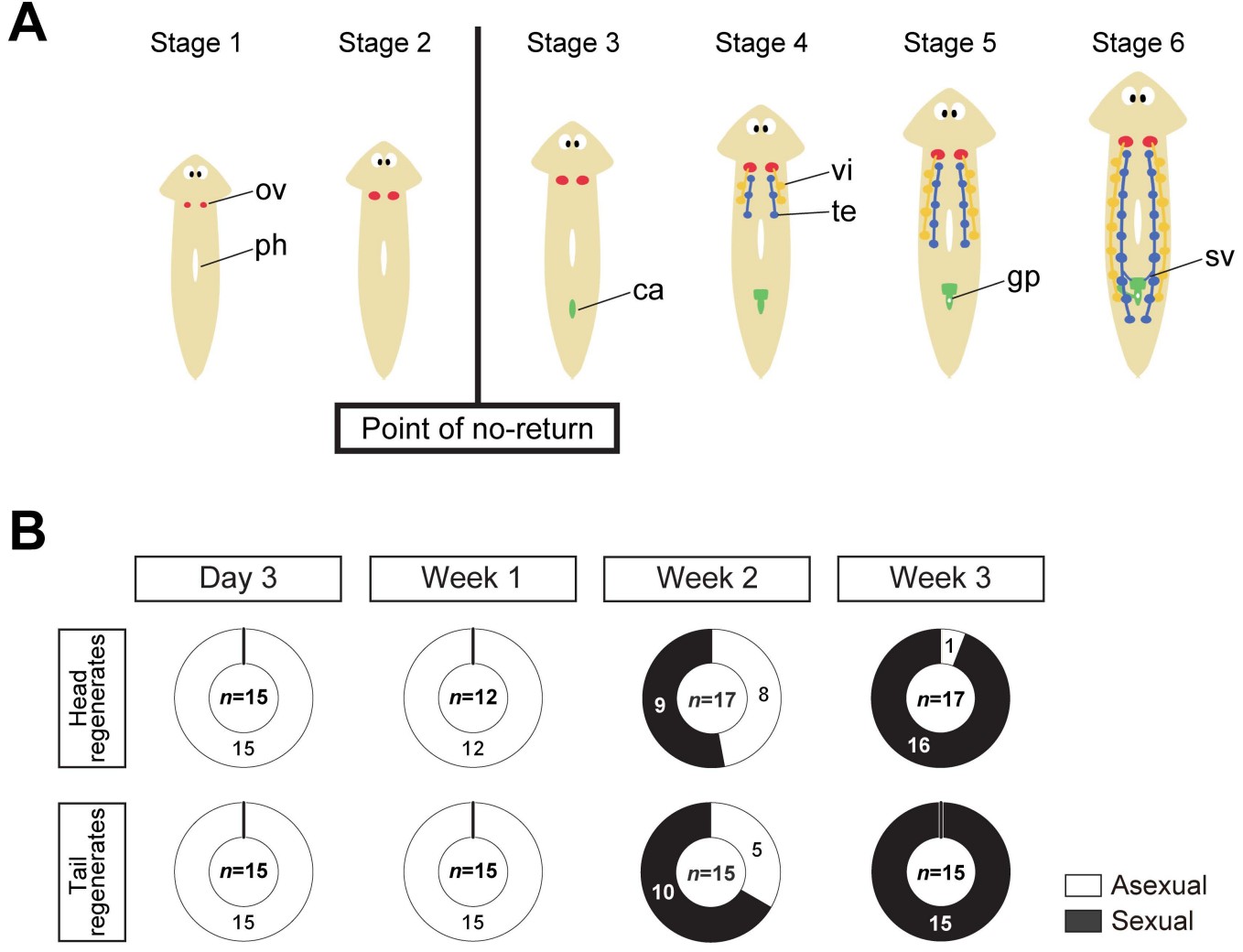

**Fig 1. Sample preparation for RNA-seq analysis at the point of no return.** (A) Six stages of sexualization in the OH strain of planarian *Dugesia ryukyuensis*. The OH worms develop hermaphroditic reproductive organs in the indicated order within approximately 1 month if they are fed daily with sexually mature worms of *Bdellocephala brunnea* or the fraction M0+M10 that contains the sex-inducing substances from *B. brunnea*. In stage 1, a pair of ovaries (ov) becomes sufficiently large to be externally visible behind the head. The ovary contains only oogonium-like cells. In stage 2, the ovary begins to mature and develop oocytes. In stage 3, a copulatory apparatus (ca) begins to form. In stage 4, the primordial testes (te) and primordial vitellaria (vi) begin to form. In stage 5, the genital pore (gp) becomes externally apparent. In stage 6, they become sexually mature with well-developed seminal vesicles (sv), ready for mating and egg-laying. ph, pharynx. Between stages 2 and 3, there is a point of no return, marking the developmental phase from which the sexualization process proceeds without further administration of sex-inducing substances. (B) The regeneration test results after feeding the Fr. M0+M10 for 3 d, 1 week, 2 weeks, or 3 weeks. After feeding with the Fr. M0+M10 that contains the sex-inducing substances derived from *B. brunnea*, the test worm was subjected to a regeneration test shown in S1 Fig. If the fragments become sexual after the regeneration test, it can be determined retrospectively that the worm had exceeded the point of no return at the time of cutting (i.e., after feeding the worm with the Fr. M0+M10 for each duration). Similarly, if the fragments become asexual, it was determined that the worm had not exceeded the point of no return.

transgressing the point of no return (Fig 1A). However, discerning the differentiation of a copulatory apparatus in stage 3 just after the point of no return from its outward appearance is difficult. Additionally, we cannot rule out the possibility that the OH worms transgress the point of no return before the differentiation of a copulatory apparatus. Thus, we conducted a "regeneration test after sexualization" to prepare worms just before and just after the point of no return for developing the RNA-seq library. If the OH worms after the point of no return are transversely cut into two pieces at the specific point

of the anterior pharynx region along the head-tail axis (prepharyngeal level), and then the head and tail fragments are allowed to regenerate, they always produce sexual worms. However, in the case of the OH worms before the point of no return, the head and tail fragments regenerate to produce only asexual worms [41,46].

Previously, we established a fractionation procedure to obtain a slightly hydrophobic fraction, namely fraction M0+M10 (Fr. M0+M10), that contains the sex-inducing substances [46]. Under this experimental procedure, the mRNA of *Bdellocephala brunnea* would be eliminated from Fr. M0+M10 through degradation and separation. The OH worms of *Dugesia ryukyuensis* were fed daily with the food containing Fr. M0+M10 in four distinct durations of the feeding (3 d, 1 week, 2 weeks, and 3 weeks). The test worms were transversely cut into two pieces at the prepharyngeal level. Total RNAs were extracted from the head fragments of half of the test worms and the tail fragments of the other half. The other fragments, from which RNA was not extracted, were used for "regeneration test after sexualization" (S1 Fig). All the test worms fed for 3 d and 1 week did not transgress the point of no return, whereas almost all the 3-week-fed test worms transgressed it. To note, approximately half of the 2-week-fed test worms transgressed the point of no return (Fig 1B). This implies that the OH worms transgress the point of no return at approximately 2 weeks of the feeding.

At each time point of sexualization, when the reproductive modes of the regenerates were identical in the regeneration test after sexualization, total RNAs extracted from the head and tail fragments of the corresponding other half were mixed (S1 Fig). The mixed RNA samples that were considered as RNAs from one individual were used for developing the RNA-seq library (S1 Table). Consequently, test worms in week 2 of sexualization were divided into two groups before and after the point of no return, namely "week 2 before-worms" and "week 2 after-worms." In addition, as a control for sexualization, OH worms were fed with chicken liver for 3 d and transversely cut into two pieces at the prepharyngeal level (S1 Table). Thus, we performed RNA-seq of the control worms, day 3-worms, week 1-worms, week 2 before-worms, week 2 after-worms, and week 3-worms (see S1 Table; three biological replicates for each worm type). On average, each sequencing library produced 11.8 M±3.95 M reads (mean±SD) after quality-control filtering, yielding 211,816,716 reads in total (S2 Table). Previously, we performed RNA-seq to produce transcriptome catalogues of asexual and sexual worms of *D. ryukyuensis* and *de novo* assembly of transcript models using all reads [47]. The reads were mapped to the transcript models with a mapping rate of >95% for all libraries (S2 Table).

## Expression analysis of candidate genes essential for transgressing the point of no return

The expression of essential genes necessary for progressing beyond the critical threshold is anticipated to be stimulated by the application of sex-inducing substances prior to reaching this point of no return. An analysis of differentially expressed genes (DEGs) in the RNA-seq data from control and week 2 before-worms was performed. In total, 443 DEGs were identified (likelihood ratio test, false discovery rate [FDR] = 0.01). In the transcript models of asexual and sexual worms of *D. ryukyuensis*, 57,762 coding DNA sequences (CDSs) were predicted, and 29,734 CDSs were annotated [47]. Of the 443 DEGs, 334 DEGs were included in this CDS list (S3 Table). We expected that the expression of essential genes required for transgressing the point of no return would be maintained or even elevated in fully sexualized worms. Using fragments per kilobase of exon per million reads mapped (FPKM) value in the RNA-seq data of asexual and sexual worms [47], we selected from the 334 DEGs those with FPKM values 10-fold higher in sexual worms than in asexual worms as the candidate genes (Fig 2A).

Experimental validation of the candidate genes using quantitative reverse-transcription polymerase chain reaction (RT-qPCR) and whole-mount *in situ* hybridization (WISH) was performed. Of the 12 candidate genes, six genes were annotated (Fig 2B). TR47468|c0_g1_i1, annotated as histone H3, was eliminated from all analyses because its function was expected to be directly unrelated to sexualization. TR79275|c0_g1_i1 was also eliminated because the short sequence length precluded primer design for RT-qPCR. All genes analyzed using RT-qPCR were expressed more than 10-fold in sexual worms than in asexual worms (Fig 3). However, the expression of TR18170|c0_g1_i1, which was annotated as glyceraldehyde-3-phosphate dehydrogenase (GAPDH), was not significantly different between asexual

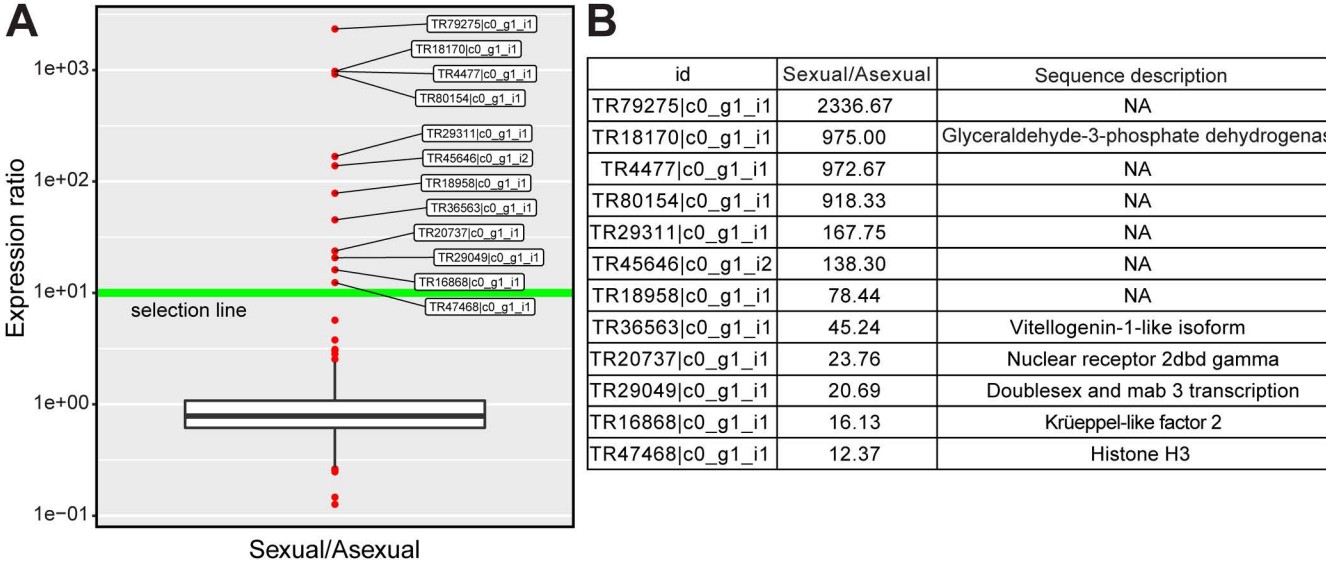

**Fig 2. Selection and annotation of candidate essential genes for sexualization.** (A) 443 DEGs between control and week 2 before-worms were obtained (FDR < 0.01). Of the 443 DEGs, 334 DEGs had CDSs. The sexual/asexual expression ratios of 334 DEGs are shown in box-and-whisker diagrams using FPKM value in the RNA-seq data of asexual and sexual worms. We selected 12 candidate genes from the 334 DEGs with FPKM values 10-fold higher in sexual worms than in asexual worms. (B) Sequence description of the candidate genes. Of the 12 candidate genes, 6 genes were annotated. NA, not applicable.

and sexual worms in WISH analysis (Fig 3A). Previously, we identified a cluster sequence DrC_01098 coding canonical GAPDH expressed strongly in asexual and sexual worms using peptide mass fingerprinting [48,49]. DrC_01098 was identical to TR24168|c0_g1_i1 in the transcriptome catalogs of asexual and sexual worms of *D. ryukyuensis*. The FPKM value of TR24168|c0_g1_i1 was approximately 10 times higher than that of TR18170|c0_g1_i1, even in sexual worms. As the template DNA sequence for the probe for *in situ* hybridization of TR18170|c0_g1_i1 (904 bp) was a 68.6% match to the corresponding sequence of TR24168|c0_g1_i1 at the nucleotide level, the ubiquitous signals common to asexual and sexual worms would recognize the expression of TR24168|c0_g1_i1. Each candidate gene other than TR18170|c0_g1_i1 demonstrated an expression pattern unique to sexual worms (Fig 3B–J). Dot-like signals on dorsal and ventral sides were detected in TR4477|c0_g1_i1 and TR29049|c0_g1_i1 (Fig 3B, 3I, high magnification of squares bounded by the yellow line). Particularly, the dot-like signals of TR29049|c0_g1_i1, annotated as double sex and mab3 transcription factor, were observed in the testicular region on the dorsal side. Except for TR29049|c0_g1_i1, the transcriptome catalogs of *D. ryukyuensis* contained four genes annotated as double sex and mab3 transcription factors. We obtained top hits using a Protein BLAST search of these DM domain genes as the query (S2A Fig). In the planarian *Schmidtea mediterranea*, the four DM domain genes, *Smed-dmd-1*, *-2*, *-3*, and *-4* were identified [50].

Maximum likelihood and neighbor-joining analyses using these genes containing splice forms in *S. mediterranea*, *doublesex* in *Drosophila melanogaster* and *male abnormal 3* in *Caenorhabditis elegans* revealed that TR29049|c0_g1_i1, TR28773|c0_g1_i2, TR38879|c0_g1_i1, and TR33587|c0_g1_i1 were categorized into clades of *Smed-dmd-1*, *-2*, *-3*, and *-4*, respectively (S2B Fig). In *S. mediterranea*, of the four DM domain genes, only *Smed-dmd-1* was expressed in the reproductive system [50]. The FPKM values of the four DM domain genes other than TR29049|c0_g1_i1 suggested that the expressions in sexual worms were not higher than those in asexual worms (S2C Fig). The expression pattern of TR29049|c0_g1_i1 was similar to that of *Smed-dmd-1* in terms of expression in the regions of testes and copulatory apparatus (Fig 3I, see Fig 1 in [50]). Based on the above, we named TR29049|c0_g1_i1 as *Dr-dmd-1*.

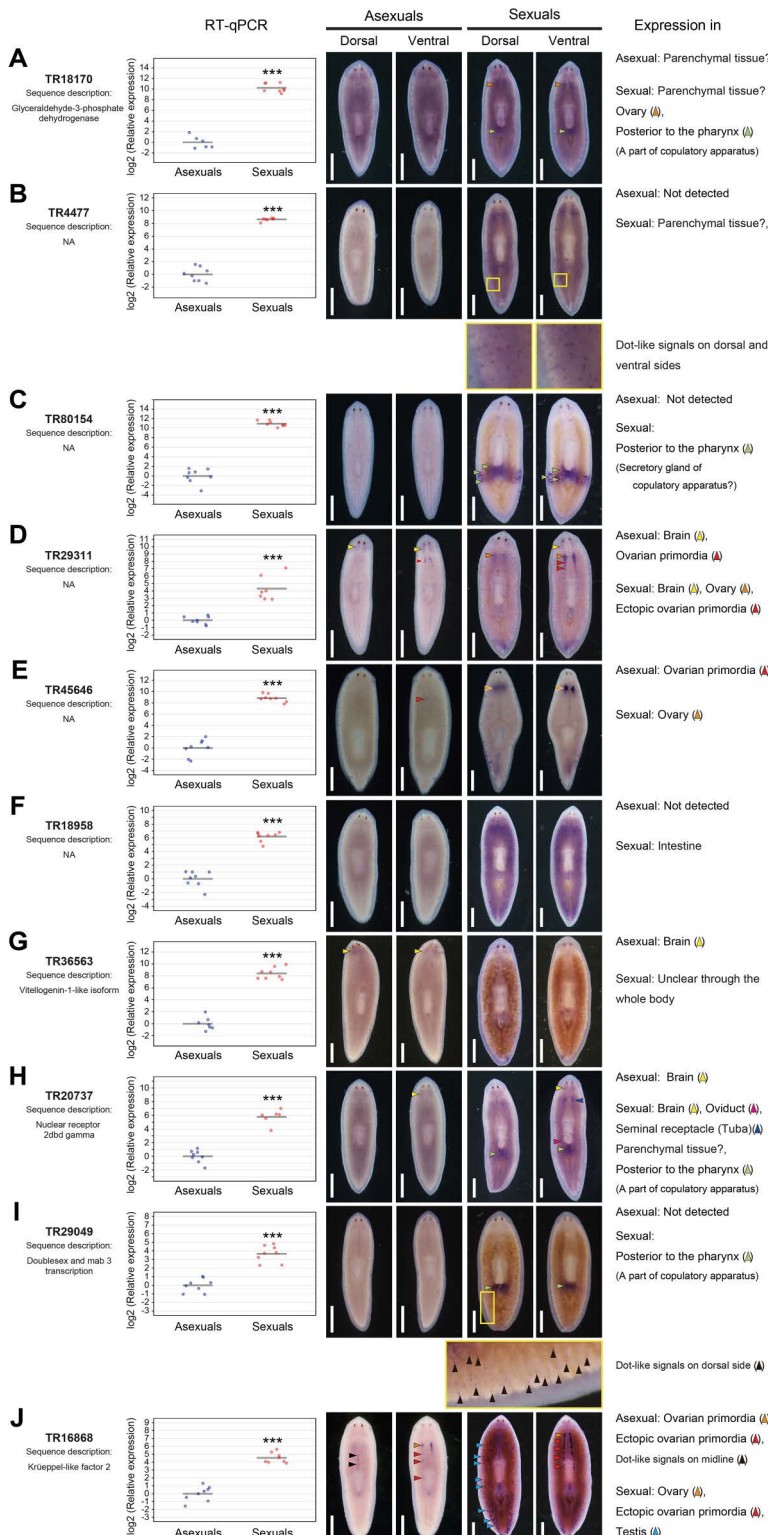

**Fig 3. Qualitative and quantitative validation of candidate essential genes for sexualization.** The expression of 10 genes among the 12 candidate genes selected in asexual and sexual worms were verified quantitatively using RT-qPCR and qualitatively using WISH; (A) TR18170|c0_g1_i1, (B) TR4477|c0_g1_i1, (C) TR80154|c0_g1_i1, (D) TR29311|c0_g1_i1, (E) TR45646|c0_g1_i2, (F) TR18958|c0_g1_i1, (G) TR36563|c0_g1_i1, (H)

TR20737|c0_g1_i1, (I) TR29049|c0_g1_i1, (J) TR16868|c0_g1_i1. The RT-qPCR data are shown relative to the expression level in the asexual worm, and log2 (relative expression) on the vertical axis indicates -ΔΔCt. The raw data are shown in S1 File. Each circle indicates an asexual or a sexual worm. Eight replicates were used, but data were handled as NA (not available) if the expression was too low to be detected or in the case of outliers (S1 File). The bars in the plots indicate the averages of -ΔΔCt. Asterisks indicate significant differences between the asexual and sexual worms (Student's or Welch's *t*-test: ***P < 0.001). Representative whole-mount *in situ* hybridization patterns for the ventral and dorsal sides of worms are shown. The expression pattern was judged based on five and three replicates in the asexual and sexual worms, respectively. Signals were seen as blue/purple staining. A scale bar, 1 mm. In TR4477|c0_g1_i1 and TR29049|c0_g1_i1, high magnifications of squares bounded by the yellow line are shown.

Next, we validated using RT-qPCR analysis whether the expressions of the candidate genes were elevated before the point of no return by stimulation of the sex-inducing substances because they were selected as DEGs between control and week 2 before-worms. As the sex-inducing activity in daily feeding with the minced worms of *B. brunnea* was almost identical to that in Fr. M0 + M10 (Figs 1B and S3), we used RNA samples derived from individuals fed daily with the minced worms for RT-qPCR analysis focusing on the point of no return. Expression of all the candidate genes was significantly elevated before the point of no return compared with that of control worms (Fig 4, a range highlighted with yellow). To note, TR20737|c0_g1_i1, annotated as nuclear receptor 2dbd gamma, was the earliest of the candidate genes to show a significant and progressive increase in expression from 3 d of sexualization onward (expression of TR18958|c0_g1_i1 also began to increase on 3 d of sexualization, but this change was not significant at one week). In *S. mediterranea*, 23 putative nuclear hormone receptor (nhr) genes were identified (see Table S1 in [51]). We performed a BLASTX search for our transcriptome catalogs by using these *nhr* genes of *S. mediterranea* as the query with an e-value cut-off of 10 (-30). TR20737|c0_g1_i1, TR52215|c0_g1_i1, TR22794|c0_g1_i1, TR44767|c0_g1_i1, TR35444|c0_g1_i1, TR3311|c0_g1_i1, TR38648|c0_g2_i1, TR22176|c0_g1_i1, TR48962|c0_g1_i1 and TR19882|c0_g1_i2 were identified as homologs of *nhr-1*, *-2*, *-6*, *-9*, *-11*, *-13*, *-14*, *-18*, *-19*, and *-20*, respectively (S4A Fig). Similar to the *nhr* genes in *S. mediterranea*, the FPKM values of the putative *nhr* genes other than TR20737|c0_g1_i1 in sexual worms were not higher than those in asexual worms (S4B Fig). The expression pattern of TR20737|c0_g1_i1 is similar to that of *nhr-1* of *S. mediterranea* (Fig 3H, see Fig 1C in [51]). Maximum likelihood and neighbor-joining analyses using the planarian *nhr* genes and the *nhr* genes from other animals revealed that *Dr-nhr-1* was categorized into a clade containing *nhr-1* of *S. mediterranea* (*Smed-nhr-1*) (S4C Fig). Based on the above, we named TR20737|c0_g1_i1 as *Dr-nhr-1*.

TR16868|c0_g1_i1, annotated as krüeppel-like factor 2 (Fig 2), was a homolog of *klf4l* (krüeppel-like factor 4-like) in *S. mediterranea* (S5A Fig). The expression pattern of TR16868|c0_g1_i1 in gonads is similar to that of *klf4l*, although TR16868|c0_g1_i1 expression was not recognized in vitellaria, unlike *S. mediterranea* (Fig 3J, see Fig 1B in [52]). We named TR16868|c0_g1_i1 as *Dr-klf4l*. Maximum likelihood and neighbor-joining analyses using the planarian *klfl* genes and the krüppel-like factor (KLF) family in human and mouse revealed that *Dr-klf4l* was categorized into a clade containing KLF17 and KLF18 (S5B Fig).

## Identification of three essential genes required for transgressing the point of no return

We conducted an RNA interference (RNAi) assay with the assay system of sexualization for the candidate genes (Fig 5A). As pretreatment of knockdown with sexualization, RNAi treatment was started from asexual worms of the OH strain. After feeding with the food containing dsRNA twice, they were transversely cut into three pieces at the prepharyngeal and postpharyngeal levels. The middle (pharyngeal) fragments were allowed to regenerate and then fed with the RNAi food three times. The head fragments and tail fragments without pharynx were not used in the feeding experiments, because they took longer to regenerate to the point where feeding was possible compared to the middle fragments. We expected that if the candidate genes are essential genes required for transgressing the point of no return, the knocked-down test worms cannot keep a sexual state without the administration of sex-inducing substances. Therefore, after the

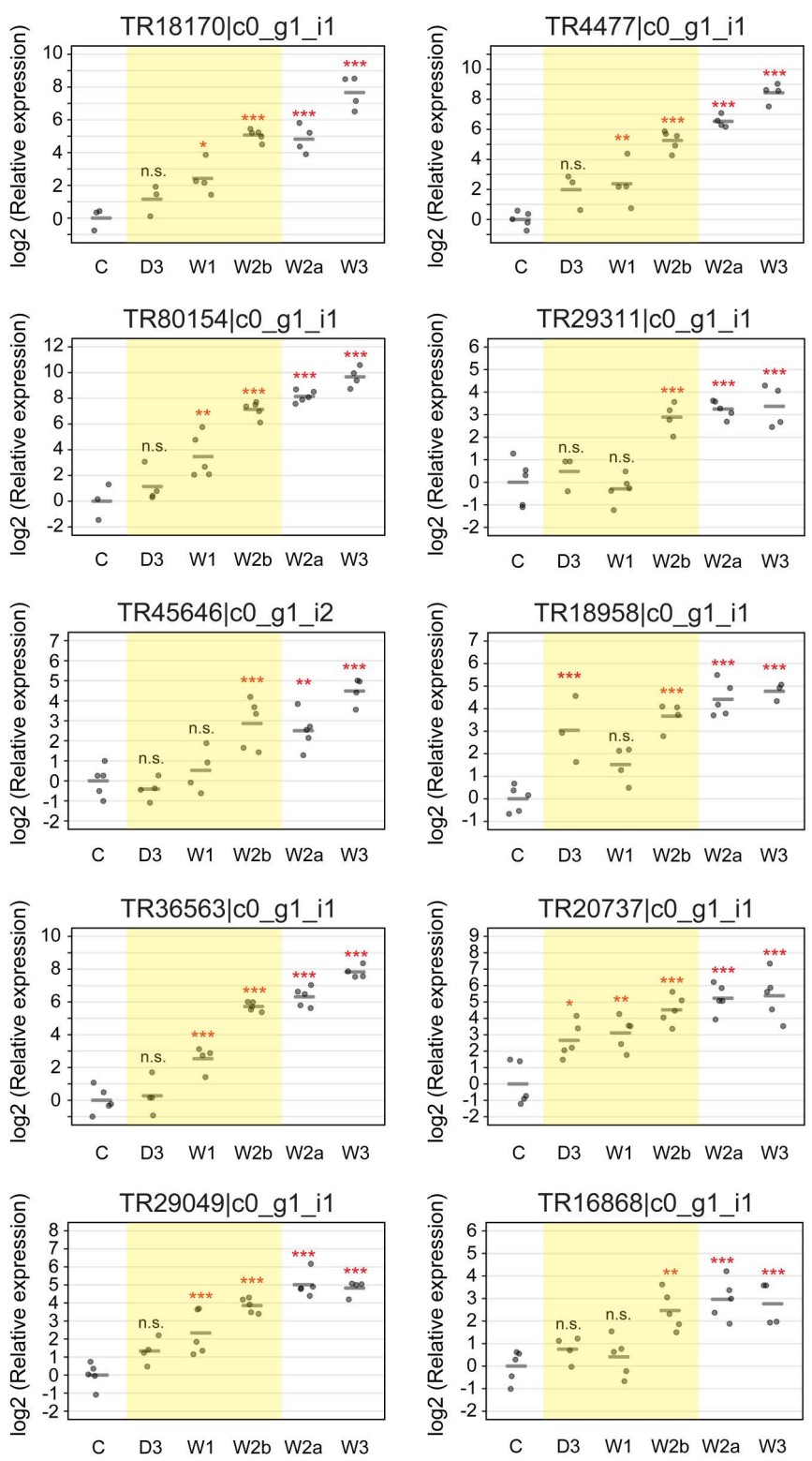

**Fig 4. Expression of candidate essential genes for sexualization significantly increased before the point of no return.** Test worms that were fed with minced *B. brunnea* and checked via regeneration on the point of no return (S1 and S3 Figs) were used as the RNA samples. C, control worms (asexual OH worms); D3, day 3-worms; W1, week 1-worms; W2b, week 2 before-worms; W2a, week 2 after-worms; W3, week 3-worms. The RT-qPCR

data for each candidate essential gene are shown relative to the expression level in the control worm, and log2 (relative expression) on the vertical axis indicates -ΔΔCt. The raw data are shown in S2 File. Each circle indicates an individual worm in the control or minced *B. brunnea*-fed groups. Five replicates were used, but data were handled as NA (not available) if the expression was too low to be detected or in the case of outliers (S2 File). The bars in the plots indicate the averages of -ΔΔCt. Asterisks indicate significant differences compared with control worms (Tukey's HSD test: *$P < 0.05$; **$P < 0.01$; ***$P < 0.001$; n.s., not significant). The results of the test worms before the point of no return, other than control worms, are shown in a range highlighted using yellow.

test worms were sexualized by feeding with minced *B. brunnea* worms mixed with dsRNA, the knocked-down test worms were transversely cut into two pieces at the prepharyngeal level and then the head fragments of half of the test worms, and the tail fragments of the other half were used for "regeneration test after sexualization" as shown in S1 Fig (Fig 5A). To assess the differentiation of reproductive organs, we performed histological examinations on two head and two tail fragments from four knocked-down worms of each candidate gene (Fig 5A and S4 Table). Remaining fragments not used in the regeneration tests were utilized for total RNA extraction. We mixed head and tail total RNAs from individuals at similar sexualization stages, resulting in eight mixed RNA samples for RT-qPCR analysis (S5 Table). The knockdown effect of dsRNA was confirmed by a decrease in gene expression (Figs 6 and S6). For evaluating morphological differences, we used RT-qPCR markers for reproductive organs: TR34905|c0_g1_i1 for ovaries [44], *DrY1* for testis [47,48,53], and *Dryg* for vitellaria [54]. Additionally, we isolated TR44991|c0_g1_i2, identified as a copulatory apparatus marker gene (S7A Fig).

After 5 weeks of sexualization, when approximately two-thirds of control worms were sexualized to stages 5–6, all *Dr-nhr-1* (RNAi) worms appeared to be asexual (Fig 5B). Histological examinations revealed that no reproductive organs appeared in *Dr-nhr-1* (RNAi) worms (Fig 5C). Consistent with the histological observations, *Dr-nhr-1* (RNAi) worms demonstrated a significantly down-regulated expression of marker genes for reproductive organs when compared to control worms (Fig 6). As expected, the regenerates of *Dr-nhr-1* (RNAi) worms became asexual, whereas the regenerates of control worms became sexual (Fig 5B).

On the contrary, although *Dr-dmd-1* (RNAi) worms were not sexualized to stage 5–6, they externally developed a copulatory apparatus that was one of the criteria for transgressing the point of no return under the administration of sex-inducing substances (Fig 5B). Histological examination of *Dr-dmd-1* (RNAi) worms revealed that ovaries and vitellaria normally developed, but the copulatory apparatus was extremely immature, and testes were not entirely recognized (Fig 5C). The results of RT-qPCR analysis in *Dr-dmd-1* (RNAi) worms were consistent with the histological observation (Fig 6). In "the regeneration test after sexualization," the regenerates of *Dr-dmd-1* (RNAi) worms could not develop the reproductive organs and eventually became asexual (Fig 5B).

Moreover, approximately half of *Dr-klf4l* (RNAi) worms were externally sexualized to stages 5–6 like control worms (Fig 5B). However, although a copulatory apparatus normally appeared to develop, ovaries were very small and of unusual morphology, and testes were not recognized (Figs 5C and 6). The only discrepancy was between the morphology and marker gene expression on vitellaria in the *Dr-klf4l* (RNAi) worms. This may be because the marker gene for vitellaria, *Dryg,* begins to be expressed before vitellaria organization occurs. In "the regeneration test after sexualization," like the regenerates of *Dr-dmd-1* (RNAi) worms, those of *Dr-klf4l* (RNAi) worms became asexual (Fig 5B).

In conclusion, the knockdown of *Dr-nhr-1, Dr-dmd-1*, or *Dr-klf4l* did not allow test worms to transgress the point of no return (Figs 5 and 6). Importantly, it should be noted that a common feature in the knockdown of *Dr-nhr-1, Dr-dmd-1*, and *Dr-klf4l* was that testicular differentiation was not observed (Figs 5 and 6). This suggests that testicular differentiation is a necessary and sufficient condition for transgressing the point of no return. The knocked-down worms of TR29311|c0_g1_i1 died due to curling and lysis phenotypes, regardless of sexualization. No phenotypes were observed in the knock-down worms of the candidate genes other than *Dr-nhr-1, Dr-dmd-1*, *Dr-klf4l*, and TR29311|c0_g1_i1 (S6 Fig).

Using RT-qPCR analysis, we confirmed that the *Dr-nhr-1* expression was not reduced in the *Dr-dmd-1* (RNAi) and *Dr-klf4l* (RNAi) worms (Fig 7A). In contrast, expressions of *Dr-dmd-1* and *Dr-klf4l* were significantly reduced in the *Dr-nhr-1*

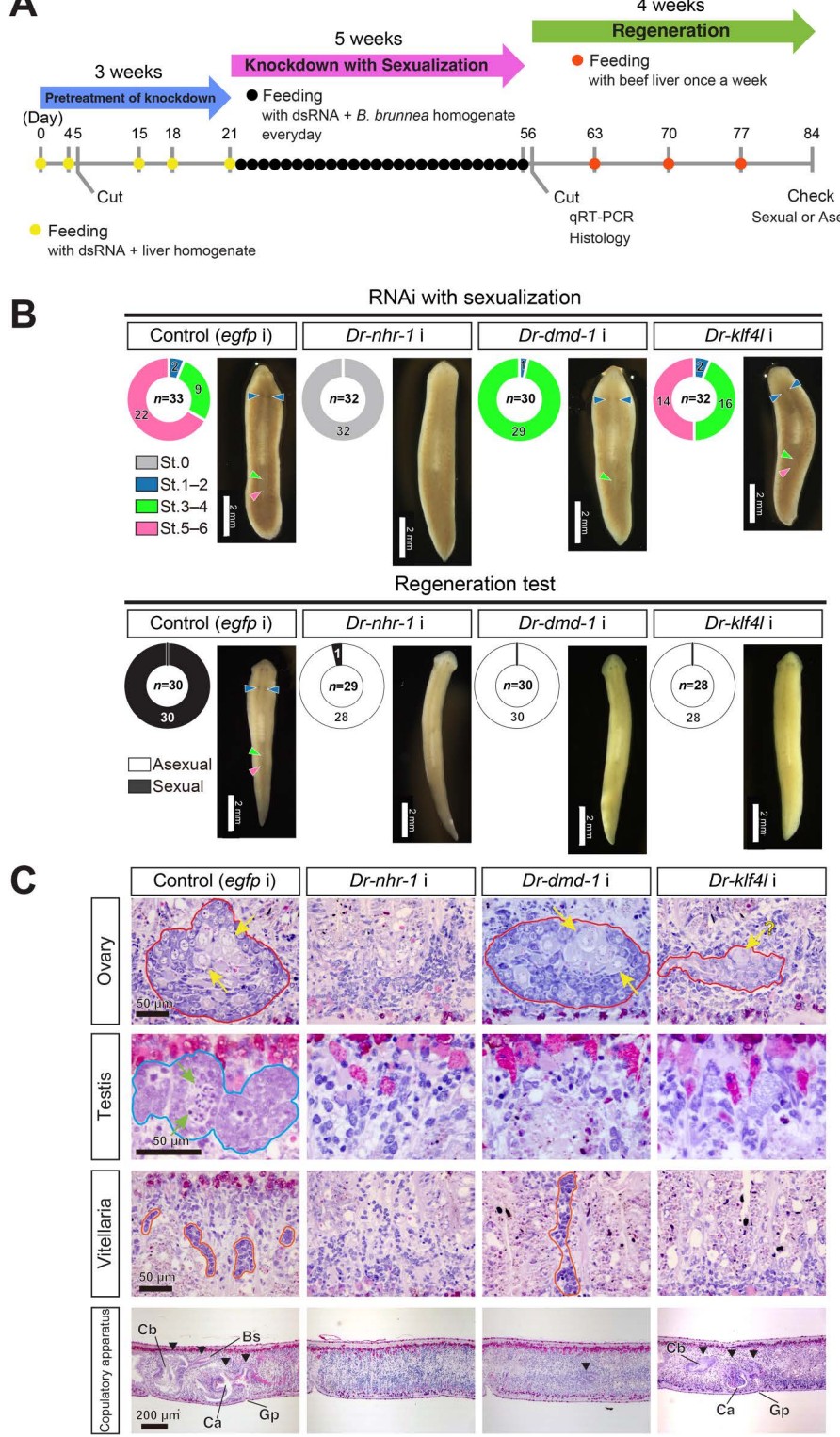

**Fig 5. Functional analysis of candidate essential genes in sexualization.** (A) Scheme of an experimental schedule of RNA interference (RNAi) with sexualization. (B) Phenotypes of RNAi gene knockdown of *Dr-nhr-1*, *Dr-dmd-1*, and *Dr-klf4l*. The worms were evaluated through external observation after 5 weeks of RNAi treatment with sexualization and after the subsequent regeneration test, and the results are shown in a donut chart with four

distinctions [Stage 0 (asexual), Stages 1–2, Stages 3–4, and Stages 5–6] and a donut chart with two distinctions (Asexual and Sexual), respectively. The number of test worms is shown in the center of the doughnut chart. The doughnut chart displays the number of worms at each of the distinctions in its circular sections. Live ventral images of the most sexually mature test worm are presented. Blue arrowheads highlight a pair of ovaries, while green and pink arrowheads point out the copulatory apparatus and genital pore, respectively. (C) Histological examination of the knocked-down worms. Representative worms were embedded, sagittally sectioned, and stained with hematoxylin and eosin (HE). The dorsal sides are at the top. Domains bound by the red and blue lines are the ovary and testis, respectively. The cell indicated using yellow and green arrows is an oocyte and a spermatid, respectively. Domains bound by the orange-colored line are developing vitellaria. A copulatory apparatus is indicated by black arrowheads, respectively. Ca, common antrum; Cb, copulatory bursa; Bs, bursa stalk; Gp, genital pore.

(RNAi) worms (Fig 7B and 7C). The respective knockdown of *Dr-dmd-1* and *Dr-klf4l* did not affect each other's expression (Fig 7B and 7C). We also examined the expression of *Dr-nanos,* which is the ortholog of *nanos* in *S. mediterranea* and is required for germ cell development [55], in the knocked-down test worms. *Dr-nanos* expression in the *Dr-dmd-1* (RNAi) worms did not differ from that of control worms, whereas *Dr-nanos* expression in the *Dr-nhr-1* (RNAi) and *Dr-klf4l* (RNAi) worms was significantly reduced compared to that of the *Dr-dmd-1* (RNAi) worms (Fig 7D).

To understand why *Dr-nanos* and *Dr-klf4l* expression, which are expressed in the testes, did not decrease in *Dr-dmd-1* (RNAi) worms despite the lack of testicular differentiation, we performed WISH analysis of *Dr-nanos*, *Dr-klf4l,* and *Dr-nhr-1* in *Dr-dmd-1* (RNAi) worms (Fig 7E). There was an important difference in the *Dr-nanos* and *Dr-klf4l* expression patterns in the dorsal regions of the control and the knocked-down worms. In the control worms, *Dr-nanos* and *Dr-klf4l* were expressed in the testes (Fig 7E, aqua blue arrowheads), whereas in the knocked-down worms, dot-like expression of *Dr-nanos* and *Dr-klf4l* was detected around the prepharyngeal region (Fig 7E, white arrowheads). Although testes have not been formed in *Dr-dmd-1* (RNAi) worms (Figs 5 and 6), *Dr-klf4l+/ Dr-nanos+* cells appear to have been induced. The *Dr-nanos* and *Dr-klf4l* expression patterns in the ovaries and supernumerary ovaries were the same in the ventral regions of the control and the knocked-down worms. There was no difference in the expression pattern of *Dr-nhr-1* between the control and the knocked-down worms.

RT-qPCR analysis revealed that the expressions of *Dr-nhr-1, Dr-dmd-1*, and *Dr-klf4l* were markedly elevated before the point of no return (Fig 4). The development of testes and copulatory apparatus is an obvious criterion beyond the point of no return (Fig 1A). We performed a WISH analysis of these essential genes in comparison with the expression pattern of testis and copulatory apparatus marker genes and *Dr-nanos* during the sexualization process (Fig 8). In this analysis, testis and copulatory apparatus marker genes TR34243|c0_g1_i1 and TR44991|c0_g1_i2, respectively, were used (S7 Fig). TR34243|c0_g1_i1 was a homolog of C3H-zinc finger-containing protein 1 in *S. mediterranea* (S7B Fig) [56]. These markers started to express around week 3 of sexualization, which was an obvious stage beyond the point of no return (Fig 8A). However, signals of the three essential genes and *Dr-nanos* began to be detected in the presumptive differentiation region of the testes or copulatory apparatus before the point of no return.

In week 1 of sexualization, dot-like signals of the three essential genes and *Dr-nanos* were arranged in a reticulate pattern in the presumptive differentiation region of the testes (Fig 8B). The *Dr-nhr-1* signals were also recognized in the presumptive differentiation region of the copulatory apparatus on the ventral side (Fig 8C). In week 2 of sexualization, the dot-like signals of *Dr-nhr-1* disappeared, whereas those of *Dr-dmd-1, Dr-klf4l*, and *Dr-nanos* were stronger (Fig 8D). In the presumptive differentiation region of the copulatory apparatus, the *Dr-nhr-1* signals were stronger, and those of *Dr-dmd-1* were also recognized. In week 3 of sexualization, the expression patterns of the dot-like signals of *Dr-dmd-1* and *Dr-klf4l* were similar in the presumptive differentiation region of testes (Fig 8E, black arrowheads), whereas a reticulate pattern of *Dr-dmd-1* signals was recognized in the periphery of the dot-like signal (Fig 8E, white arrowheads). Moreover, the *Dr-nhr-1* signals around the copulatory apparatus were recognized in seminiferous tubules and oviducts, whereas the signals of *Dr-dmd-1* were recognized in seminiferous tubules and sperm ducts (Fig 8F). In week 4 of sexualization, in testes, the signals of *Dr-dmd-1* were almost not detected, whereas both signals of *Dr-klf4l and Dr-nanos* appeared to overlap (Fig 8G). Eventually, in sexual worms, *Dr-dmd-1* expression in the mesenchymal space disappeared, leaving only a dot-like signal (Fig 3I).

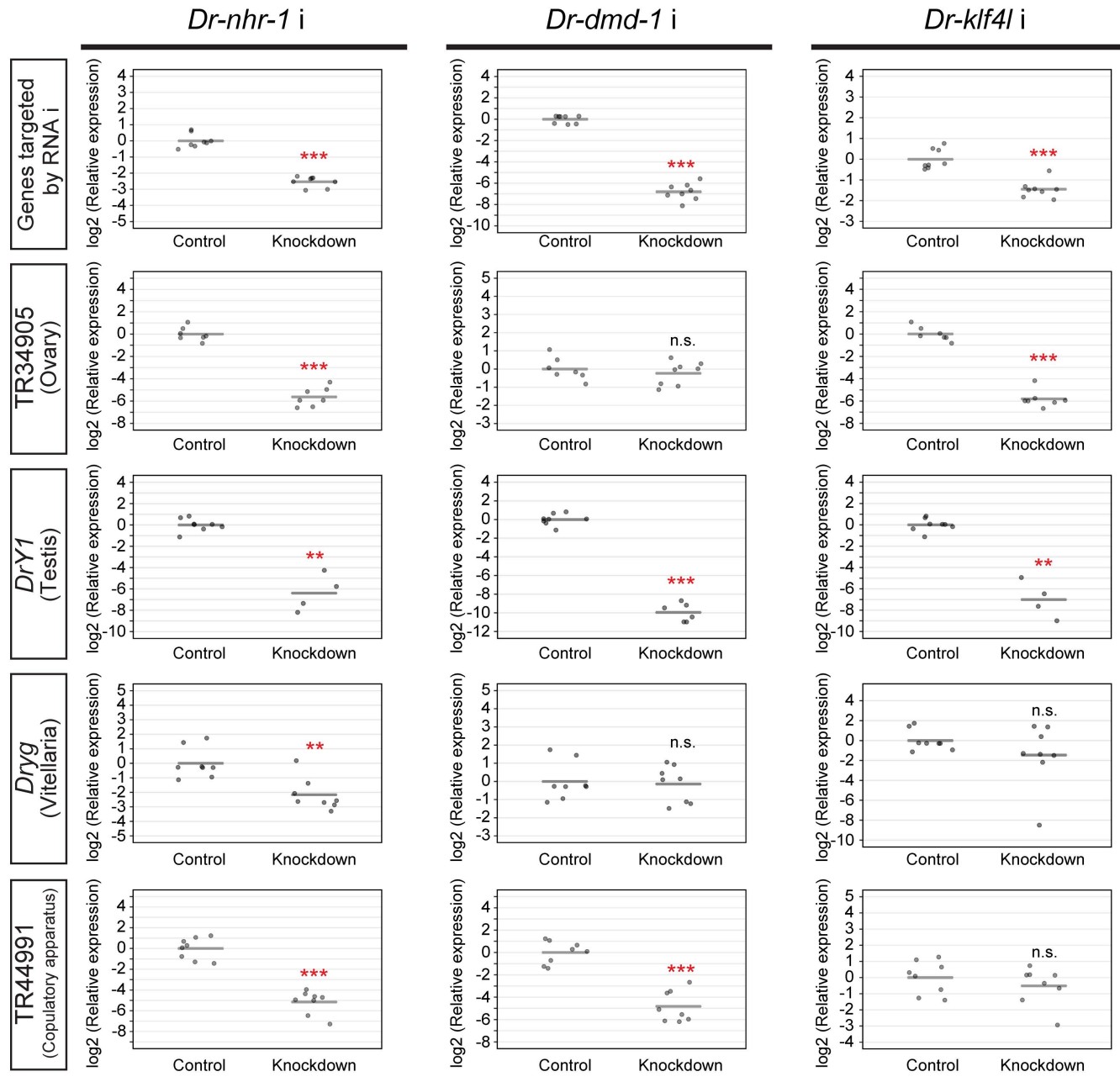

**Fig 6. Quantitative validation of sexualization in the knocked-down worms.** The top row shows the RNAi efficiency of *Dr-nhr-1, Dr-dmd-1,* and *Dr-klf4l*, respectively. The bottom three rows show the differentiation levels of the ovaries, testes, vitellaria, and copulatory apparatus in the knocked-down worms of *Dr-nhr-1, Dr-dmd-1,* and *Dr-klf4l*. The RT-qPCR data are shown relative to the expression level in the control worm, and log2 (relative expression) on the vertical axis indicates -ΔΔCt. The raw data are shown in S3 File. Each circle indicates a control or a knocked-down worm. Eight replicates were used, but data were handled as NA (not available) if the expression was in the case of outliers (S3 File). The bars in the plots indicate the averages of -ΔΔCt. Asterisks indicate significant differences between the control and knocked-down worms (Student's or Welch's *t*-test: **P<0.01; ***P<0.001; n.s., not significant).

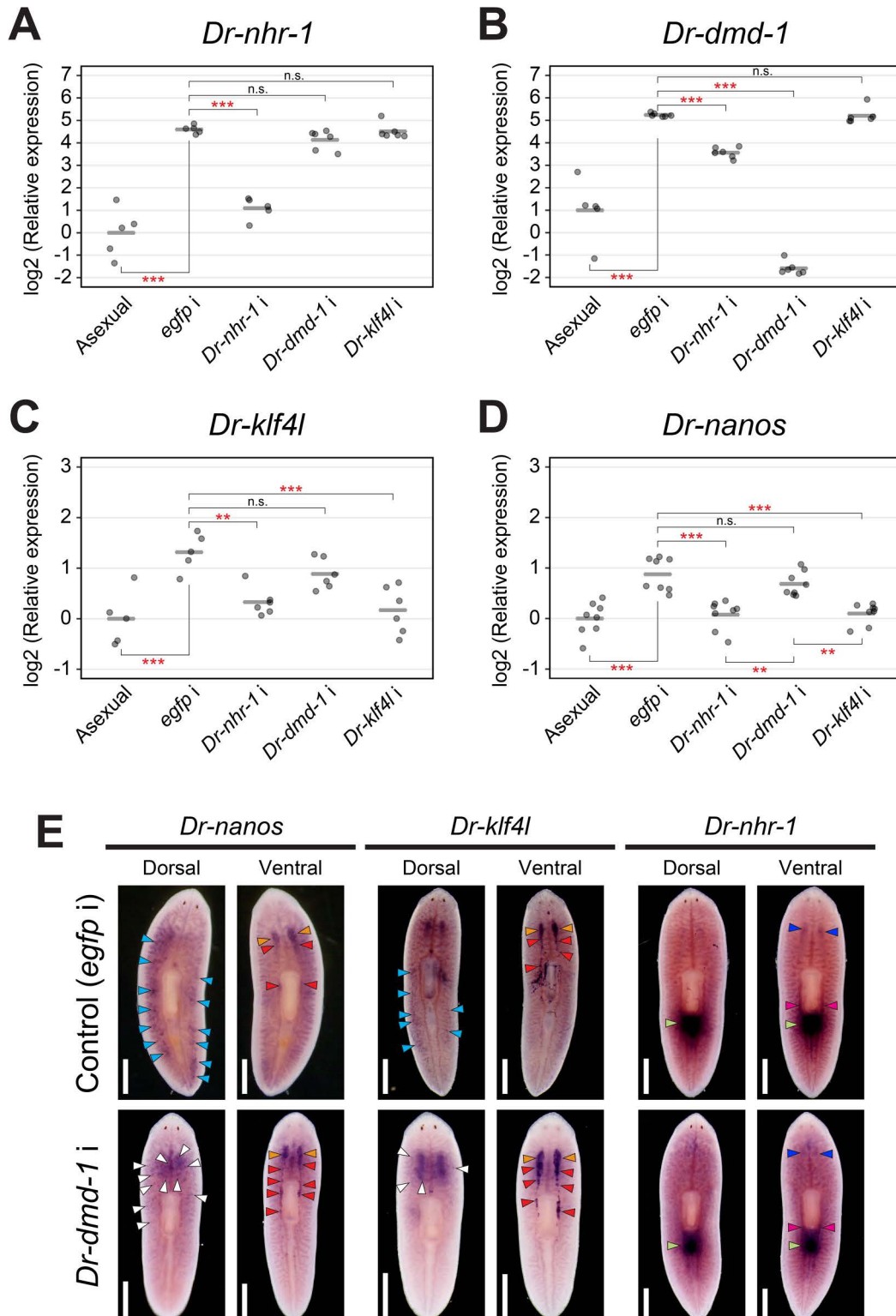

**Fig 7. Estimation of hierarchical relationship among the three essential genes in sexualization using RT-qPCR.** The RT-qPCR data for the three essential genes, (A) *Dr-nhr-1,* (B) *Dr-dmd-1*, (C) *Dr-klf4l*, and (D) an ortholog of the *nanos* gene, *Dr-nanos,* are shown relative to the expression level in asexual worms, and log2 (relative expression) on the vertical axis indicates -ΔΔCt. The raw data are shown in S4 File. Each circle indicates an asexual

worm, an *egfp* i control worm, or the RNAi knocked-down worm of *Dr-nhr-1, Dr-dmd-1*, or *Dr-klf4l* (*Dr-nhr-1* i, *Dr-dmd-1* i, or *Dr-klf4l* i). Six replicates for the expression of *Dr-nhr-1, Dr-dmd-1*, and *Dr-klf4l*, and eight replicates for *Dr-nanos* expression were used, but data were handled as NA (not available) if the expression was too low to be detected or in the case of outliers (S4 File). The bars in the plots indicate the averages of -ΔΔCt. Asterisks indicate significant differences compared with the asexual worms or the *egfp* i control worms (Tukey's HSD test: **P < 0.01; ***P < 0.001; n.s., not significant). (E) *Dr-dmd-1* (RNAi) worms were used for whole-mount *in situ* hybridization samples of *Dr-nanos, Dr-klf4l*, and *Dr-nhr-1*. The expression pattern was judged based on 5–6 replicates. Signals were seen as blue/purple staining. A scale bar, 1mm. Ovaries, supernumerary ovaries, seminal receptacle (tuba), oviduct, a copulatory apparatus, testis, and a dot-like signal in the prepharyngeal region are indicated by orange, red, blue, magenta, green, aqua blue, and white arrowheads, respectively.

## Finding of *Dugesia ryukyuensis*-sexual induction-related innexin (*Dr-siri*)

To find genes involved in testicular differentiation in association with *Dr-nhr-1, Dr-dmd-1*, and *Dr-klf4l*, we performed RNA-seq analysis of the knocked-down test worms of these three essential genes (S4 and S6 Tables; five biological replicates for each worm type). On average, each sequencing library produced 4.5 M ± 1.3 M reads (mean ± SD) after quality-control filtering, yielding 222,857,784 reads in total. The reads obtained in this study were mapped to the transcript models [47] with a mapping rate of >98% for all libraries.

An analysis of DEGs between control worms [*egfp* (RNAi) worms] and test worms with knocked-down levels of three essential genes [*Dr-nhr-1* (RNAi), *Dr-dmd-1* (RNAi), and *Dr-klf4l* (RNAi)] was performed (likelihood ratio test, FDR < 0.05). A total of 1904 DEGs (109 upregulated; 1795 downregulated), 1546 DEGs (117 upregulated; 1429 downregulated), and 1501 DEGs (107 upregulated; 1394 downregulated) were identified between control and worms with knocked-down *Dr-nhr-1* (RNAi), *Dr-dmd-1* (RNAi), and *Dr-klf4l* (RNAi), respectively. Overall, 796 DEGs (4 upregulated; 792 downregulated) common to the three test worm groups were shown using Venn diagrams (Fig 9A). In the RNA-seq library, week 3 worms, at the point of no return, which corresponds to stage 4 where testicular primordia emerge (Figs 1A and S7B), were analyzed. We found 384 DEGs between asexual and week 3 worms (likelihood ratio test, FDR < 0.05). Of the 792 downregulated genes shared among the knocked-down worms, 27 also appeared among the DEGs between control and week 3 worms (Fig 9B).

Among the 27 DEGs, we focused on TR37455|c0_g1_i1, annotated as innexin (Table 1). Innexin proteins form gap junctions between adjacent cells in invertebrates [57]. The gap junctions formed by innexins comprise adjacent intracellular hexameric hemichannels that dock to form small molecule exchange channels. Innexins are four-pass transmembrane proteins with intracellular C- and N-termini domains [58–61]. In the transcriptome catalogs of *D. ryukyuensis*, we found 46 contigs annotated as innexin with the four transmembrane domains, including TR37455|c0_g1_i1 (S7 Table). In *D. japonica*, a closely related species of *D. ryukyuensis*, 11 innexin genes (*inx2–5, inx7–13*) were identified, and their expression patterns were analyzed using WISH [62]. We performed a local BLASTX search for our transcriptome catalogs by using these innexin genes of *D. japonica* as the query. Of the 46 contigs annotated as innexin of *D. ryukyuensis*, we found the homologous genes (*Dr-inx2–5, Dr-inx7–13*) of 11 innexin genes in *D. japonica* (S7 Table). We performed alignment of the predicted amino acid sequences of TR37455|c0_g1_i1 with these innexin genes. The conserved four transmembrane domains, cysteine residues in the extracellular loops, and tetrapeptide sequence (YYQW in the second transmembrane domain) were also found in TR37455|c0_g1_i1 (S8 Fig). Phylogenic analysis of the planarian *Dugesia* innexin sequences was carried out with innexin sequences of *C. elegans* and *D. melanogaster* (S9 Fig). The innexins in *D. japonica* fall into three groups (Group I: Intestine, Group II: Nervous system/blastema, and Group III: Mesenchyme/protonephrida) according to both sequence and expression patterns [62]. Taken together, TR37455|c0_g1_i1 was classified into Group III of planarians with the addition of a new innexin gene.

RT-qPCR and WISH analyses revealed that TR37455|c0_g1_i1 was expressed in the testicular region of sexual worms, and the expression was not recognized in the WISH of asexual worms (Fig 10A and 10B). During weeks 1–2 of sexualization, dot-like signals of TR37455|c0_g1_i1 were recognized in a reticulate pattern in the presumptive differentiation region of testes, like those of *Dr-dmd-1* and *Dr-klf4l* (Fig 10C and 10D, black arrowheads). When the OH worms

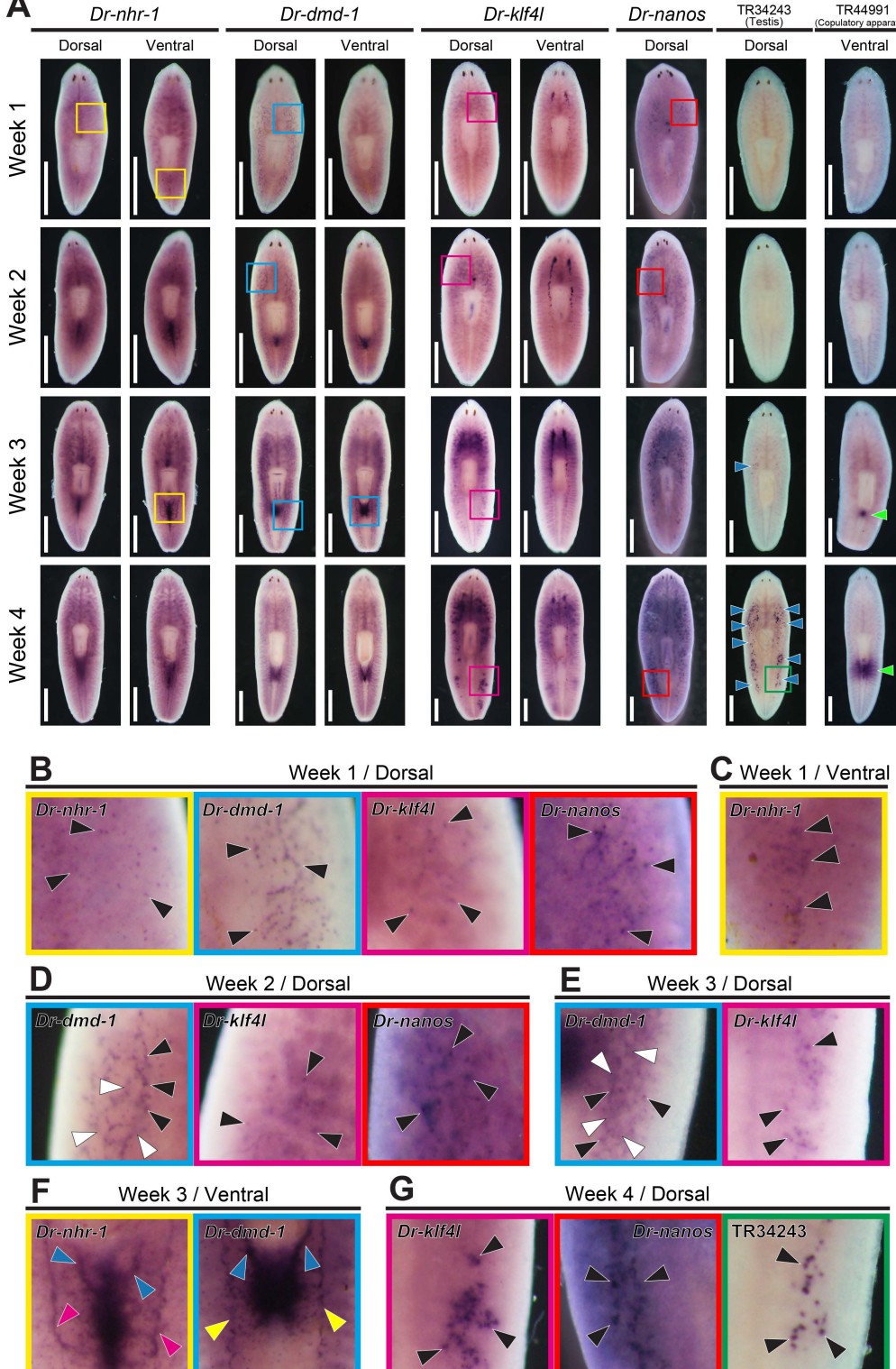

**Fig 8. Changes in expression patterns of the three essential genes during the sexualization process.** The OH worms were fed daily with minced worms of *B. brunnea* in four distinct durations of the feeding (1 week, 2 weeks, 3 weeks, and 4 weeks) and were used for samples of whole-mount *in situ* hybridization. The expression pattern was judged based on five replicates in the individuals of each week. Signals were seen as blue/purple staining. (A) Changes in the *Dr-nhr-1*, *Dr-dmd-1*, and *Dr-klf4l* expressions during the sexualization process are shown in comparison with the expression patterns

of testis and copulatory apparatus marker genes and *Dr-nanos*. The testis and the copulatory apparatus are indicated with blue and green arrowheads, respectively. A scale bar, 1 mm. (B) In presumptive differentiation region of testes on the dorsal side in week 1 of sexualization in (A), high magnification views of the area denoted with yellow (*Dr-nhr-1*), blue (*Dr-dmd-1*), magenta (*Dr-klf4l*), and red (*Dr-nanos*) boxes are shown. Dot-like signals of *Dr-nhr-1, Dr-dmd-1, Dr-klf4l*, and *Dr-nanos* are indicated using black arrowheads. (C) In the presumptive differentiation region of the copulatory apparatus on the ventral side in week 1 of sexualization in (A), a high magnification view of the area denoted by the yellow (*Dr-nhr-1*) box is shown. The signals of *Dr-nhr-1* are indicated by black arrowheads. (D–E) In the presumptive differentiation region of testes on the dorsal side in weeks 2 and 3 of sexualization in (A), high magnification views of the areas denoted by the blue (*Dr-dmd-1*), magenta (*Dr-klf4l*), and red (*Dr-nanos*) boxes are shown. Dot-like signals of *Dr-dmd-1, Dr-klf4l*, and *Dr-nanos* are indicated using black arrowheads. A reticulate pattern of *Dr-dmd-1* signals in the periphery of the dot-like signals is also recognized (white arrowheads). (F) In the differentiation area of the copulatory apparatus on the ventral side in week 3 of sexualization in (A), high magnification views of the area denoted with yellow (*Dr-nhr-1*) and blue (*Dr-dmd-1*) boxes are shown. Seminal vesicles and oviducts in which *Dr-nhr-1* is expressed are indicated with blue and magenta arrowheads, respectively. Seminal vesicles and sperm ducts in which *Dr-dmd-1* is expressed are indicated with blue and yellow arrowheads, respectively. Oviducts, seminal vesicles, and spermatic ducts were determined by their location and shape based on the expression patterns of *nhr-1* and *Smed-dmd-1* in *S. mediterranea* [50,51]. (G) In testes on the dorsal side in week 4 of sexualization in (A), high magnification views of the area denoted with green (a testis marker gene, TR34243|c0_g1_i1), magenta (*Dr-klf4l*), and red (*Dr-nanos*) boxes are shown. Signals of TR34243|c0_g1_i1, *Dr-klf4l,* and *Dr-nanos* are indicated by black arrowheads.

were fed daily with minced worms of *B. brunnea,* testicular primordia appeared around week 3 of sexualization (Fig 8A). In week 3 of sexualization, the signals were stronger in the differentiation region of the testes (Fig 10D, black arrowheads). In the differentiation region of testes in week 4 of sexualization, the strong signals of TR37455|c0_g1_i1 were observed in the cell mass-like structures (Fig 10D, black arrowheads), whereas a reticulate pattern of TR37455|c0_g1_i1 signals was also recognized in the periphery of the cell mass-like structures (Fig 10D, white arrowheads). At four weeks of sexual differentiation, the testes exhibit dorsal development; however, the vitellaria remain undeveloped at this stage. Therefore, the observed cell mass with a prominent signal is anticipated to correspond to the testes. In sexual worms, TR37455|c0_g1_i1 appears to be expressed not only in the testes but throughout the mesenchymal cells surrounding the testes in the dorsal region (Fig 10B). Of note, the TR37455|c0_g1_i1 expression pattern in sexual worms, especially in the mesenchymal space, was very similar to the expression pattern of *Dr-dmd-1* in week 3 of sexualization (Fig 8E). Furthermore, RT-qPCR analysis revealed that the expression of TR37455|c0_g1_i1 was significantly elevated from week 1 of sexualization (Fig 10E, a range highlighted with yellow) and that the expressions of TR37455|c0_g1_i1 were significantly decreased in the knocked-down worms of the three essential genes (Fig 10F). It was noted that TR37455|c0_g1_i1 expression in *Dr-nhr-1* (RNAi) and *Dr-dmd-1* (RNAi) worms was considerably lower than that in *Dr-klf4l* (RNAi) worms.

As the expression analysis results implied that TR37455|c0_g1_i1 is also required for transgressing the point of no return, we conducted an RNAi experiment with the assay system of sexualization (Figs 5A and 11 and S8 Table). The phenotype of TR37455|c0_g1_i1 (RNAi) worms was remarkably similar to that of *Dr-dmd-1* (RNAi) worms (Figs 5–6, and 11). Histological examination and RT-qPCR analysis revealed that TR37455|c0_g1_i1 (RNAi) worms normally developed ovaries and vitellaria, but not testes (Fig 11B, 11C and S9 Table). The failure of testicular differentiation is consistent with the expression pattern of TR37455|c0_g1_i1 in the differentiation region of the testes. TR37455|c0_g1_i1 was not expressed in the copulatory apparatus but was slightly more immature in TR37455|c0_g1_i1 (RNAi) worms than in control worms. In the "regeneration test for sexualization," the regenerates derived from all TR37455|c0_g1_i1 (RNAi) worms became asexual, indicating that they did not transgress the point of no return (Fig 11A). Based on the above, we named TR37455|c0_g1_i1 *Dr-siri* (*Dugesia ryukyuensis-*sexual induction-related innexin).

### *Dr-siri* was expressed not only in testes but also in dorsal mesenchymal cells

As in WISH analysis, the expression of *Dr-siri* seemed to overlap with the expressions of *Dr-klf4l*, *Dr-nanos*, and *Dr-dmd-1* in the testicular region (Figs 3, 8, and 10), we performed double staining using fluorescence *in situ* hybridization (FISH). Unfortunately, test worms fed with *B. brunnea* during the sexualization process were not suitable for FISH analysis due to the high background fluorescence. Thus, we used sexual worms for FISH analysis in this study.

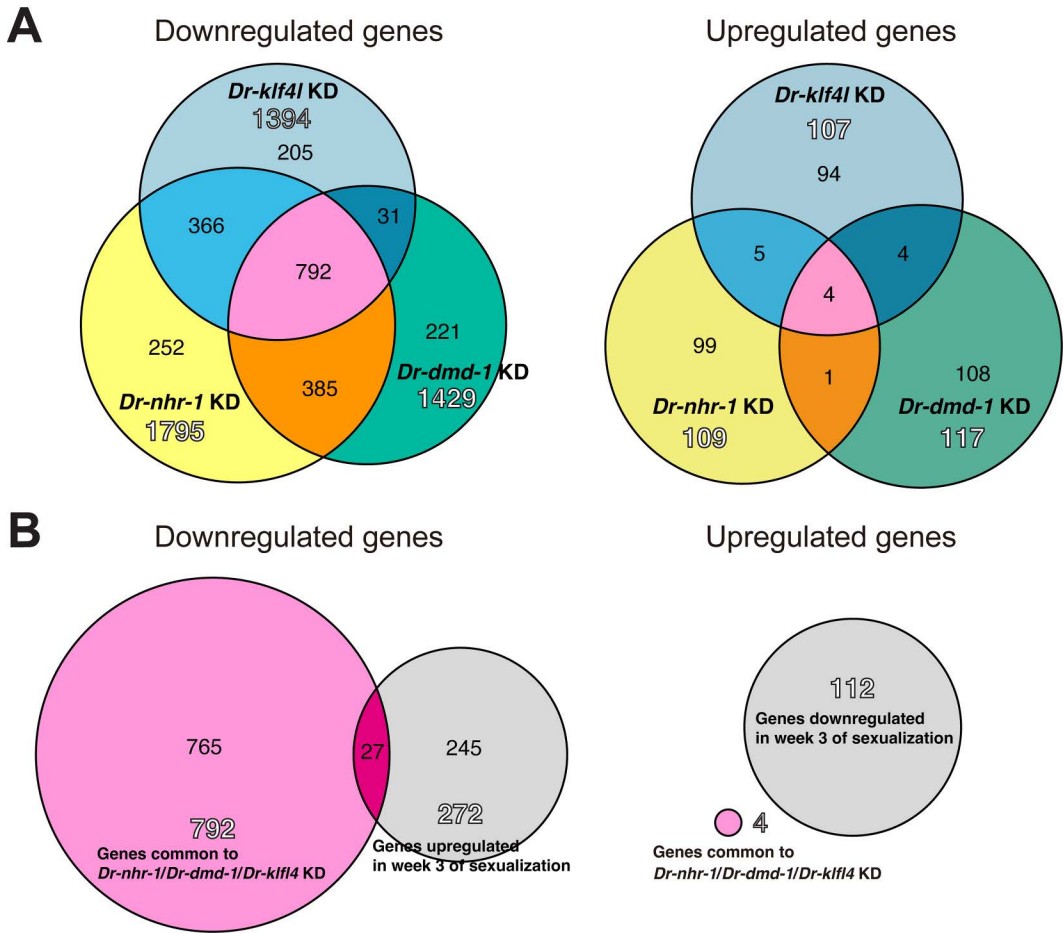

**Fig 9. Selection of candidate genes involved in testicular differentiation.** (A) Venn diagrams are used to guide the selection of genes common to the knockdown of *Dr-nhr-1*, *Dr-dmd-1*, and *Dr-klf4l*. A common feature in the knockdown of *Dr-nhr-1*, *Dr-dmd-1*, and *Dr-klf4l* was a failure of testicular differentiation. We expected that genes responsible for testicular differentiation would be included in genes common to the knockdown of *Dr-nhr-1*, *Dr-dmd-1*, and *Dr-klf4l*. DEGs that are downregulated and upregulated in the knockdown of each gene were analyzed separately. The outline number indicates the number of DEGs in the knockdown of each gene. In total, 796 DEGs (792 downregulated genes, 4 upregulated genes) common to the knockdown of *Dr-nhr-1*, *Dr-dmd-1*, and *Dr-klf4l* are shown in pink using Venn diagrams. (B) Venn diagrams are used to guide the selection of candidate genes responsible for testicular differentiation. As testicular primordia emerged in week 3 of sexualization, it was expected that 384 DEGs between asexual and week 3-worms include genes responsible for early testicular differentiation. The genes upregulated and downregulated for testicular differentiation should be downregulated or upregulated in the genes common to the knockdown of *Dr-nhr-1*, *Dr-dmd-1*, and *Dr-klf4l*, respectively. Thus, we performed a Venn diagram approach using the downregulated and upregulated genes common to the knockdown of *Dr-nhr-1*, *Dr-dmd-1*, and *Dr-klf4l*, and the genes upregulated and downregulated in week 3-worms. Of the 792 downregulated genes common to the knockdown of *Dr-nhr-1*, *Dr-dmd-1*, and *Dr-klf4l*, 27 genes were included in 272 genes upregulated in week 3-worms.

Firstly, we performed paraffin section FISH (Fig 12). Planarian *klf4l* and *nanos* are expressed in germline cells, including gamete stem cells (GSCs) in the testis [52,55,63,64]. Like *S. mediterranea*, *Dr-klf4l* + / *Dr-nanos* + cells (Fig 12A, yellow arrowheads) and *Dr-klf4l* –/ *Dr-nanos* + cells (Fig 12A, white arrowheads) were recognized in the testis. In the double staining using FISH on *Dr-nanos* or *Dr-klf4l* and *Dr-siri*, cells in which *Dr-nanos* and *Dr-siri*, or *Dr-klf4l* and *Dr-siri* co-expressed, cells in which *Dr-nanos* or *Dr-klf4l* expressed alone, and cells in which *Dr-siri* expressed alone were observed in the testis (Fig 12B and 12C). *dmd-1* in *S. mediterranea* is expressed in the niche cells of the testis [50]. However, the expression of *Dr-dmd-1* in the testes of sexually mature worms was very weak (Fig 3I). In the double staining using FISH on *Dr-klf4l* and *Dr-dmd-1*, the signal of *Dr-dmd-1* was not detected (Fig 12D), whereas in

**Table 1. Annotation of candidate genes responsible for testicular differentiation.**

| id | Sexual/Asexual | Sequence description |
|---|---|---|
| TR4477\|c0_g1_i1 | 9788 | NA** |
| TR67454\|c0_g1_i1 | 7257.731959 | High mobility group protein dsp1 |
| TR31264\|c0_g1_i1 | 6551.077788 | Tubulin polymerization-promoting protein family |
| TR79276\|c0_g1_i1 | 6509.202454 | Sjchgc04698 protein |
| TR38673\|c1_g1_i2 | 5458.764727 | Y-box-binding protein 3 |
| TR8063\|c0_g1_i1 | 5347.56461 | NA |
| TR29450\|c0_g2_i1 | 5319.148936 | Phosphoenolpyruvate cytosolic |
| TR24479\|c0_g1_i1 | 5207.667732 | NA |
| TR34243\|c0_g1_i1 | 5069.008783 | Zinc finger CCCH domain-containing protein 31 |
| TR10202\|c0_g2_i1 | 5057.636888 | Migration and invasion enhancer 1 |
| TR35334\|c0_g1_i1 | 4841.155235 | Plastin 3 |
| TR55411\|c0_g1_i1 | 4624 | Carrier protein mitochondrial-like |
| TR25623\|c1_g1_i1 | 4210.353866 | Expressed conserved protein |
| TR5979\|c0_g2_i1 | 4118.811881 | Spermatid elongation defective 1 |
| TR956\|c0_g1_i1 | 3678.16092 | NA |
| TR79459\|c0_g1_i1 | 3628.753412 | NA |
| TR6058\|c0_g1_i1 | 3443.199297 | NA |
| TR73220\|c0_g1_i1 | 3392.42685 | Elav-like protein 3 isoform x8 |
| TR26521\|c0_g1_i1 | 3144.288577 | Spermatid elongation defective 1 |
| TR5998\|c0_g1_i1 | 2848.729792 | 78 kDa glucose-regulated protein |
| TR48084\|c4_g1_i1 | 2773.685547 | Tubulin alpha chain |
| TR61226\|c0_g1_i1 | 1784.697807 | NA |
| TR18170\|c0_g1_i1 | 1552.554507 | Glyceraldehyde-3-phosphate dehydrogenase |
| TR27221\|c0_g1_i1 | 1014.469453 | Homeobox protein xhox-3-like |
| TR37455\|c0_g1_i1 | 270.4481793 | Innexin unc-9 |
| TR48084\|c2_g1_i1* | 201.7128691 | Sodium bicarbonate transporter-like protein 11 isoform x1 |
|  |  | Tubulin alpha-1a chain-like |
| TR42323\|c0_g1_i3 | 3.793894477 | Caveolin-1-like |

*: This contig contained two CDSs. **: NA indicates "not applicable".

that on *Dr-nanos* and *Dr-dmd-1*, no cells were observed in which the signals from the two genes coincided, though the detection of *Dr-dmd-1* was a rare occurrence (Fig 12E). In the double staining using FISH on *Dr-dmd-1* and *Dr-siri*, no obvious signals of *Dr-dmd-1* were observed.

Since *Dr-dmd-1* expression was difficult to detect using the paraffin section FISH method, we also performed whole-mount FISH (Fig 13). *Dr-siri* was co-expressed with *Dr-klf4l* and *Dr-nanos* (Fig 13A and 13B; yellow arrowheads), while *Dr-siri* alone is expressed in the mesenchymal cells around the testes (Figs 13A, 13B and 13C; white arrowheads and S10). In the double staining using FISH on *Dr-dmd-1* and *Dr-siri*, no obvious signals of *Dr-dmd-1* were observed by even the whole-mount FISH method (Fig 13C).

## Expression pattern of the essential genes required for sexualization in starved sexual worms

We have identified four genes essential for crossing the point of no return, but it is unclear whether these genes are involved in maintaining the acquired sex. To answer this question, we focused on starved sexual worms. Planarians can

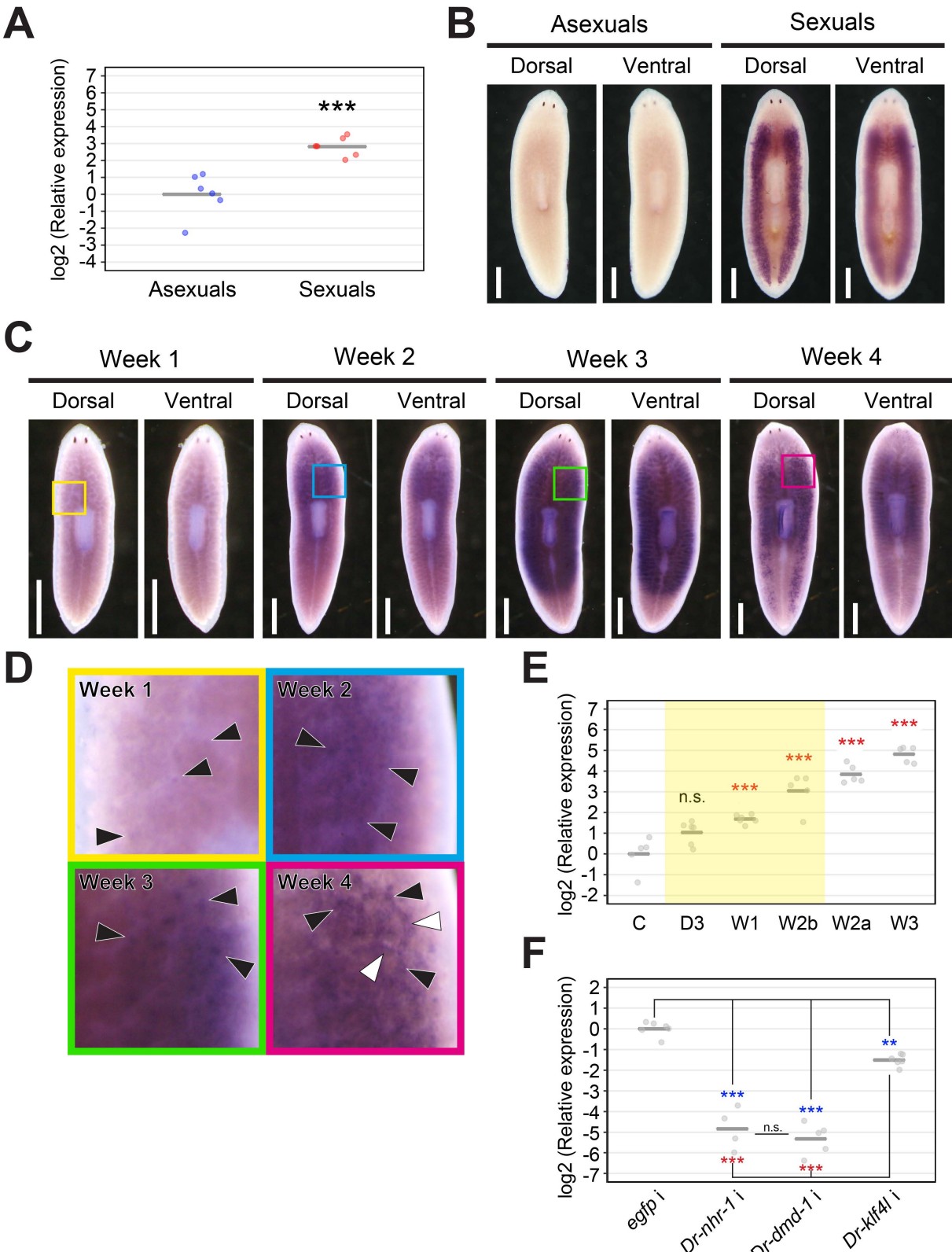

**Fig 10. Qualitative and quantitative validation of TR37455|c0_g1_i1.** (A–B) Expression of TR37455|c0_g1_i1 in asexual and sexual worms. (A) The RT-qPCR data are shown relative to the expression level in the asexual worm, and log2 (relative expression) on the vertical axis indicates -ΔΔCt. The

raw data are shown in S5 File. Each circle indicates an asexual or a sexual worm. Six replicates were used. The bars in the plots indicate the averages of -ΔΔCt. Asterisks indicate significant differences between the asexual and sexual worms (Student's *t*-test: \*\*\*P < 0.001). (B) Whole-mount *in situ* hybridization patterns of TR37455|c0_g1_i1 for the ventral and dorsal sides of worms are shown. The expression pattern was judged based on five and three replicates in the asexual and sexual worms, respectively. Signals were seen as red/purple staining. A scale bar, 1 mm. (C–D) Changes in TR37455|c0_g1_i1 expression during the sexualization process. (C) The OH worms were fed daily with minced worms of *B. brunnea* in four distinct durations of the feeding (1 week, 2 weeks, 3 weeks, and 4 weeks) and were used for samples of whole-mount *in situ* hybridization. The expression pattern was judged based on five replicates in the individuals of each week. Signals were seen as red/purple staining. A scale bar, 1 mm. (D) In (presumptive) differentiation region of testes in (C), high magnification views of the area denoted using yellow (week 1), blue (week 2), green (week 3), and magenta (week 4) boxes are shown. Dot-like signals of TR37455|c0_g1_i1 are indicated using black arrowheads. In week 4, a reticulate pattern of signals in the periphery of the dot-like signals is also recognized (white arrowheads). (E) Changes in the TR37455|c0_g1_i1 expression level during sexualization. C, control worms (asexual OH worms); D3, day 3-worms; W1, week 1-worms; W2b, week 2 before-worms; W2a, week 2 after-worms; W3, week 3-worms. The RT-qPCR datum is shown relative to the expression level in the control worm, and log2 (relative expression) on the vertical axis indicates -ΔΔCt. The raw data are shown in S5 File. Each circle indicates an individual worm in the control or minced *B. brunnea*-fed groups. Six replicates were used, but data were handled as NA (not available) if the expression was too low to be detected or in the case of outliers (S5 File). The bars in the plots indicate the averages of -ΔΔCt. Asterisks indicate significant differences compared with control worms (Tukey's HSD test: \*\*\*P < 0.001; n.s., not significant). The results of test worms, before reaching the point of no return, other than control worms, are shown in a range highlighted with yellow. (F) TR37455|c0_g1_i1 expression in the knocked-down worms of *Dr-nhr-1*, *Dr-dmd-1*, and *Dr-klf4l*. The RT-qPCR data are shown relative to the expression level in *egfp* i control worms, and log2 (relative expression) on the vertical axis indicates -ΔΔCt. The raw data are shown in S5 File. Each circle indicates the *egfp* i control worms or the knocked-down worms of *Dr-nhr-1*, *Dr-dmd-1*, or *Dr-klf4l* (*Dr-nhr-1* i, *Dr-dmd-1* i, or *Dr-klf4l* i). Six replicates for *Dr-nhr-1*, *Dr-dmd-1*, and *Dr-klf4l* expressions, but data were handled as NA (not available) if the expression was too low to be detected or in the case of outliers (S5 File). The bars in the plots indicate the averages of -ΔΔCt. Asterisks indicate significant differences compared with the *egfp* i control worms or the *Dr-klf4l* (RNAi) worms (Tukey's HSD test: \*\*P < 0.01; \*\*\*P < 0.001; n.s., not significant).

undergo "degrowth" under starvation because their body size is homeostatically regulated by "cell turnover" from aPSCs [65]. We prepared worms approximately 3–5 mm long through extreme starvation from 15–20 mm long, fully sexualized worms. The starved worms lacked visible ovaries or copulatory apparatus and appeared asexual, yet retained their acquired sex. Once their nutrition improved, they developed reproductive organs without any sex-inducing substances. Previously, we identified *Dr-nanos* + cells in the testicular region of the starved worms [66].

In this study, we performed a WISH analysis of the four essential genes and the vitellaria differentiation marker *Dryg*, in addition to *Dr-nanos,* which is expressed in testes, in the starved sexual worms (Fig 14). The expression patterns of these genes in asexual worms were shown as controls (Fig 14A–C and 14G–I). Expression of *Dr-nanos* and *Dr-klf4l* was observed in the ovarian primordia of asexual worms (Fig 14A-v and 14B-v), but no expression was detectable in the presumptive differentiation regions of other reproductive organs. In contrast, the expression of *Dr-nanos* and *Dr-klf4l* in the ovarian differentiation region was stronger in starved sexual worms than in asexual worms (Fig 14D-v and 14E-v). In addition, *Dr-nhr-1* and *Dr-dmd-1* were found to be expressed in the differentiation region of a copulatory apparatus in starved sexual worms (Fig 14F-v and 14J-v). In the testicular differentiation region (testes and the surrounding mesenchymal tissue) of starved sexual worms, we identified dot-like signals of *Dr-nanos, Dr-klf4l, and Dr-dmd-1* (Fig 14D-d, 14E-d, and 14J-d) but not the signal of *Dr-siri* (Fig 14K). Particularly, a reticulate pattern of *Dr-dmd-1* signals was observed in the periphery of the dot-like signal (Fig 14J', white arrowheads). In the vitellaria differentiation region of starved sexual worms, no expression of the vitellaria marker gene *Dryg* was observed (Fig 14L).

We also conducted histological observations of five starved sexual worms and two starved asexual worms. In the starved asexual worms, only primordial ovaries were recognized (Fig 15A–C). However, in the starved sexual worms, stage 2-like ovaries and stage 3-like testes (testicular primordia) were identified (Fig 15D and 15E). Retracting copulatory organs were observed in two of the five worms (Fig 15F) but not in the remaining three worms. Even primordia of vitellaria were not observed. Taken together with the WISH results, it is expected that the dot-like signals of *Dr-klf4l* and *Dr-nanos* are expressed in the testicular primordia, and the dot-like signals of *Dr-dmd-1* and its surrounding signals are expressed in the testicular primordia and peripheral mesenchymal cells, respectively. *Dr-nhr-1* and *Dr-dmd-1* are also likely to be expressed in the mesenchymal cells (or retracting tissue) in the differentiation regions of the copulatory apparatus. It is noteworthy that *Dr-dmd-1*, which was almost no longer expressed in the testicular differentiation region of mature sexual

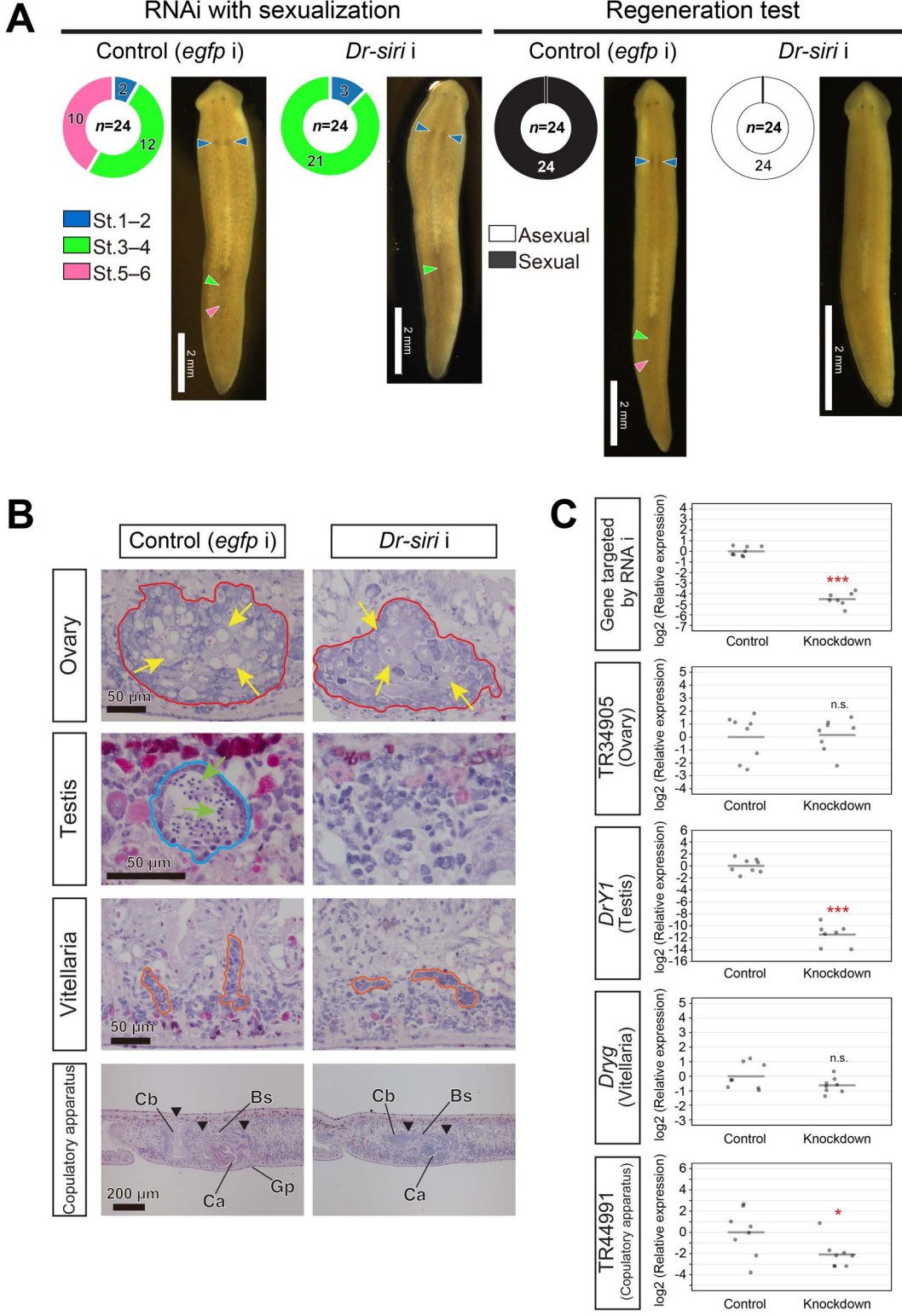

**Fig 11. Functional analysis of *Dr-siri* (TR37455|c0_g1_i1).** Based on the results shown in this figure, we named TR37455|c0_g1_i1 *Dr-siri* (*Dugesia ryukyuensis*-sexual induction-related innexin). (A) Phenotypes of RNAi gene knockdown of *Dr-siri*. The worms were evaluated through external observation after 5 weeks of RNAi treatment with sexualization and after the subsequent regeneration test, and the results are shown in a donut chart with four

distinctions [Stage 0 (asexual), Stages 1–2, Stages 3–4, and Stages 5–6] and a donut chart with two distinctions (Asexual and Sexual), respectively. The number of test worms is shown in the center of the doughnut chart. The doughnut chart displays the number of worms at each of the distinctions in its circular sections. Live ventral images of the most sexually mature test worm are presented. Blue arrowheads highlight a pair of ovaries, while green and pink arrowheads point out the copulatory apparatus and genital pore, respectively. (B) In the histological examination, representative worms were embedded, sagittally sectioned, and stained with hematoxylin and eosin (HE). The dorsal sides are located at the top. The regions delineated by the red and blue lines correspond to the ovary and testis, respectively. The cell indicated by the yellow arrow is an oocyte, while the cell marked by the green arrow is a spermatid. The areas bordered by the orange line represent the developing vitellaria. A copulatory apparatus is indicated by black arrowheads. Ca, common antrum; Cb, copulatory bursa; Bs, bursa stalk; Gp, genital pore. (C) RNAi efficiency and degree of differentiation of reproductive organs in the *Dr-siri* knocked-down worms were examined using RT-qPCR. The RT-qPCR data are shown relative to the expression level in the control worm, and log2 (relative expression) on the vertical axis indicates -ΔΔCt. The raw data are shown in S6 File. Each circle indicates a control or a knocked-down worm. Eight replicates were used, but data were handled as NA (not available) if the expression was in outliers (S6 File). The bars in the plots indicate the averages of -ΔΔCt. Asterisks indicate significant differences between the control and knocked-down worms (Student's *t*-test: *P<0.05; ***P<0.001; n.s., not significant).

worms, was again strongly expressed in starved sexual worms, and conversely, the *Dr-siri* signal, which was expressed in mature sexual worms, was no longer observed in starved sexual worms.

## Discussion

Some species of planarians can switch between vegetative and sexual reproduction depending on the environmental conditions. Individuals undergoing vegetative reproduction do not develop mature reproductive organs despite possessing adult pluripotent stem cells (an asexual state), whereas those undergoing sexual reproduction keep developing reproductive organs and mate (a sexual state). In this study, using a bioassay system of sexualization in a clonal strain of *Dugesia ryukyuensis*, we identified four essential genes involved in testicular differentiation that are necessary for terminating the asexual state. This indicates that testes production might be vital for switching from vegetative to sexual reproduction, similar to the shift from parthenogenesis to sexual reproduction. Our findings could provide a foundation for elucidating the mechanisms behind transitioning between asexual and sexual reproductive modes in metazoans.

### Three essential genes required for sexualization are transcriptional factors

We conducted RNAi knockdown during sexualization for approximately 10 candidate genes selected using RNA-seq analysis (Fig 2), resulting in identification of three essential genes required for transgressing the point of no return, the homologs of *nhr-1* (a nuclear hormone receptor gene) [51], *Smed-dmd-1* (a doublesex/male-abnormal-3 domain gene) [50] and *klf4l* (krüeppel-like factor 4-like) [52] in the planarian *Schmidtea mediterranea*, termed *Dr-nhr-1*, *Dr-dmd-1*, and *Dr-klf4l*, respectively (S2, S4, and S5 Figs). The temporal and spatial expression patterns during the sexual development of *Dr-nhr-1*, *Dr-dmd-1*, and *Dr-klf4l* (Figs 3H, 3I, 3J, and 8) were slightly different from those of each homolog in *S. mediterranea*. Moreover, the phenotypes in knockdowns of *Dr-nhr-1*, *Dr-dmd-1,* and *Dr-klf4l* (Figs 5 and 6) did not completely match those of each homolog in *S. mediterranea* [50–52]. This difference may be due to the species, namely the difference between *D. ryukyuensis*, in which sexual development, including germ cell formation, occurs postembryonically in adults in response to a change in the reproductive modes, and *S. mediterranea*, in which sexual development, including germ cell formation, already begins during embryonic to hatchling stages. For instance, the expression timing of *nanos* orthologs in the presumptive differentiation region of testes is significantly different between *S. mediterranea* and *D. ryukyuensis* [55,64,67]. The hatched juveniles of the sexual strain of *S. mediterranea* spontaneously begin to express *nanos* ortholog. Even worms of the asexual strain of *S. mediterranea* develop *nanos*-labeled cell masses in the presumptive differentiation region of testes, which are likely to be testicular primordia. In contrast, the asexual worms of the OH strain do not express *nanos* ortholog in the presumptive differentiation region of testes unless there is stimulation of the sex-inducing substances, and the testicular primordia finally appear in stage 4 of sexualization (Fig 1A). These facts are likely to cause slight differences in the expression patterns and phenotypes in knockdowns among *Dr-nhr-1*, *Dr-dmd-1*, and *Dr-klf4l* in *D. ryukyuensis* and each homolog in *S. mediterranea*. Despite the slight differences, based on the expression

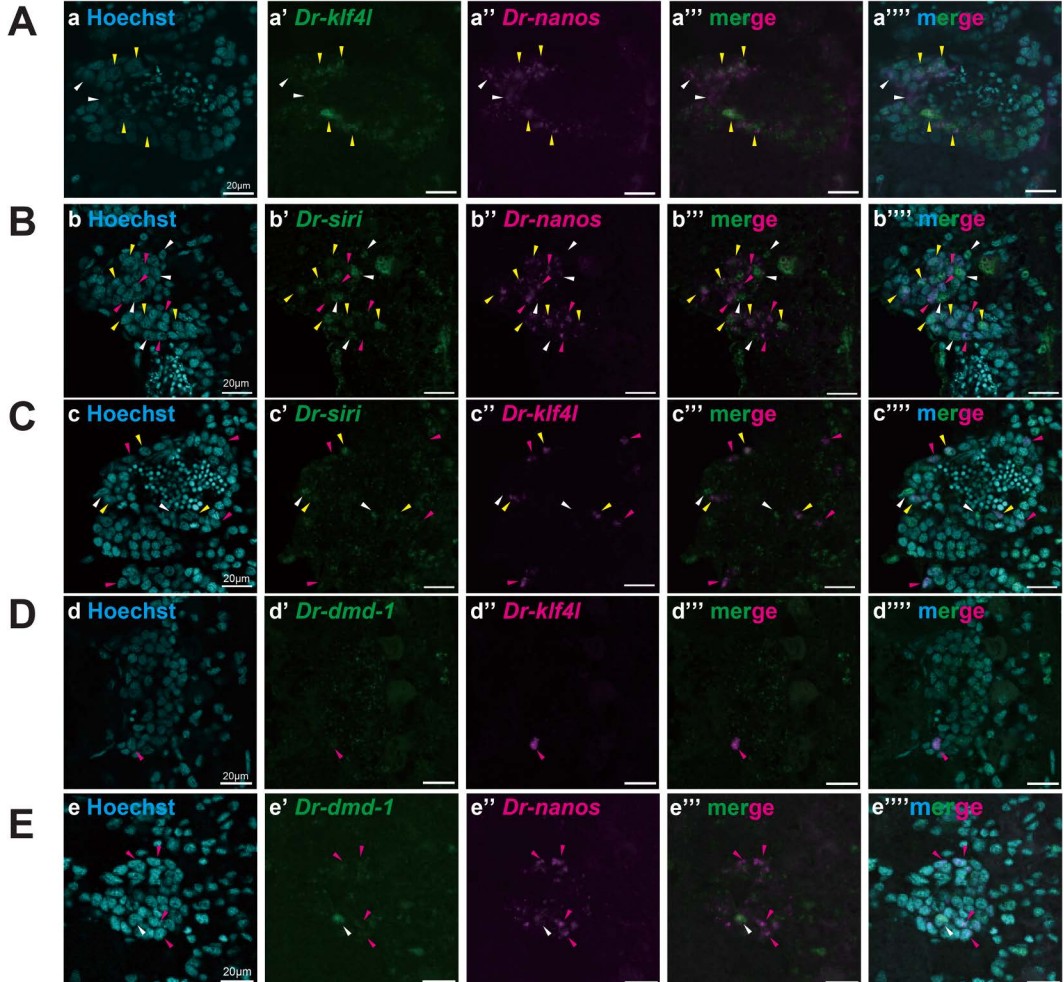

**Fig 12. _Dr-siri_ is not only co-expressed with _Dr-nanos_ and _Dr-klf4_ but also expressed alone in testes.** Confocal images showing two-color FISH for _Dr-siri_ and germ cell/niche cell gene mRNAs in the paraffin sections of sexual worms. (A) (a) The nuclei were counterstained with Hoechst 33342 (blue). (a') _Dr-klf4l_ (green), (a") _Dr-nanos_ (magenta), (a''') green-magenta merged, and (a'''') all merged images are shown. Yellow arrowheads indicate cells with overlapping expression of _Dr-nanos_ and _Dr-klf4l_. White arrowheads indicate cells expressing _Dr-nanos_ alone. (B) (b) The nuclei were counterstained with Hoechst 33342 (blue). (b') _Dr-siri_ (green), (b") _Dr-nanos_ (magenta), (b''') green-magenta merged, and (b'''') all merged images are shown. Yellow arrowheads indicate cells with overlapping expression of _Dr-nanos_ and _Dr-siri_. Magenta and white arrowheads indicate cells expressing _Dr-nanos_ and _Dr-siri_ alone, respectively. (C) (c) The nuclei were counterstained with Hoechst 33342 (blue). (c') _Dr-siri_ (green), (c") _Dr-klf4l_ (magenta), (c''') green-magenta merged and (c'''') all merged images are shown. Yellow arrowheads indicate cells with overlapping expression of _Dr-klf4l_ and _Dr-siri_. Magenta and white arrowheads indicate cells expressing _Dr-klf4l_ and _Dr-siri_ alone, respectively. (D) (d) The nuclei were counterstained with Hoechst 33342 (blue). (d') _Dr-dmd-1_ (green), (d") _Dr-klf4l_ (magenta), (d''') green-magenta merged, and (d'''') all merged images are shown. A magenta arrowhead indicates a cell expressing _Dr-klf4l_ alone. (E) (e) The nuclei were counterstained with Hoechst 33342 (blue). (e') _Dr-dmd-1_ (green), (e") _Dr-nanos_ (magenta), (e''') green-magenta merged, and (e'''') all merged images are shown. Magenta and white arrowheads indicate cells expressing _Dr-nanos_ and _Dr-dmd-1_ alone, respectively.

patterns and sequence similarities, we concluded that _Dr-nhr-1_, _Dr-dmd-1_, and _Dr-klf4l_ are orthologs of transcriptional factors _nhr-1_, _Smed-dmd-1_, and _klf4l_ in _S. mediterranea_, respectively.

### _Dr-nhr-1_ is the most upstream critical gene induced by sex-inducing substances

The knocked-down worms of _Dr-nhr-1, Dr-dmd-1_, and _Dr-klf4l_ could not maintain a sexual state (Figs 5 and 6). Notably, under the stimulation of sex-inducing substances, the differentiation of reproductive organs in the knocked-down worms

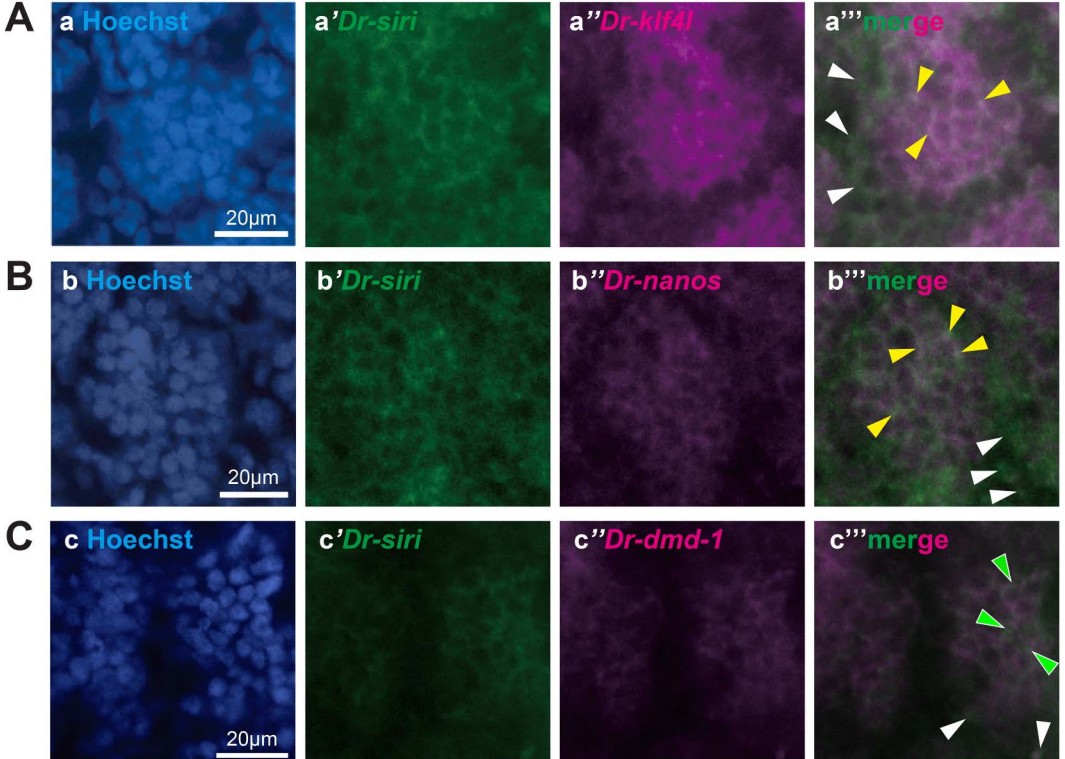

**Fig 13. *Dr-siri* is expressed not only in testes but also in dorsal mesenchymal cells.** Single confocal sections showing two-color whole-mount FISH for *Dr-siri* (green) and germ cell/niche cell gene (magenta) mRNAs in the testicular region of sexual worms. (A) (a) The nuclei were counterstained with Hoechst 33342 (blue). (a') *Dr-siri* (green), (a") *Dr-klf4l* (magenta), and (a"') green-magenta merged images are shown. (B) (b) Hoechst 33342 (blue). (b') *Dr-siri* (green), (b") *Dr-nanos* (magenta), and (b"') green-magenta merged images are shown. (C) (c) Hoechst 33342 (blue). (C') *Dr-siri* (green), (c") *Dr-dmd-1* (magenta), and (c"') green-magenta merged images are shown. Yellow arrowheads indicate cells with overlapping expression of *Dr-siri* and *Dr-klf4l* or *Dr-nano*. Green and white arrowheads indicate cells expressing *Dr-siri* alone inside and outside the testis, respectively.

was extremely different for each gene. *Dr-nhr-1* (RNAi) worms did not develop reproductive organs at all, even with stimulation using sex-inducing substances (Figs 5 and 6). *Dr-nhr-1* expression was most quickly and significantly elevated during the sexualization process (Fig 4) and was not reduced in *Dr-dmd-1 and Dr-klf4l* (RNAi) worms (Fig 7A). This suggests that *Dr-nhr-1* is the most upstream important gene required for sexualization (Fig 16). Nuclear hormone receptors are ligand-binding transcription factors, and the ligands are usually lipophilic low-molecular-weight compounds [68]. The nuclear hormone receptor encoded by planarian *nhr-1* is an orphan receptor. The sex-inducing substances suggested are slightly hydrophobic, low-molecular-weight compounds [46]. We speculated that the sex-inducing substances may be a ligand of the nuclear hormone receptor encoded by *Dr-nhr-1* or may induce ligand production as well as *Dr-nhr-1* expression.

## Predicted relationship among the three essential genes for testicular differentiation

This study revealed that in week 1 of sexualization (the period before the point of no return), dot-like signals of *Dr-nanos*, a *nanos* ortholog, as well as the three essential genes, were arranged in a reticulate pattern in the presumptive differentiation region of testes (Fig 8B). Notably, the dot-like signals of *Dr-nhr-1* disappeared since week 2 of sexualization. The gonads of animals contain two types of tissues: the germ cell lineage that gives rise to gametes and the somatic cells that give rise to all other tissues that support and maintain gametogenesis. Similar to Sertoli cells in mammalian testes, the

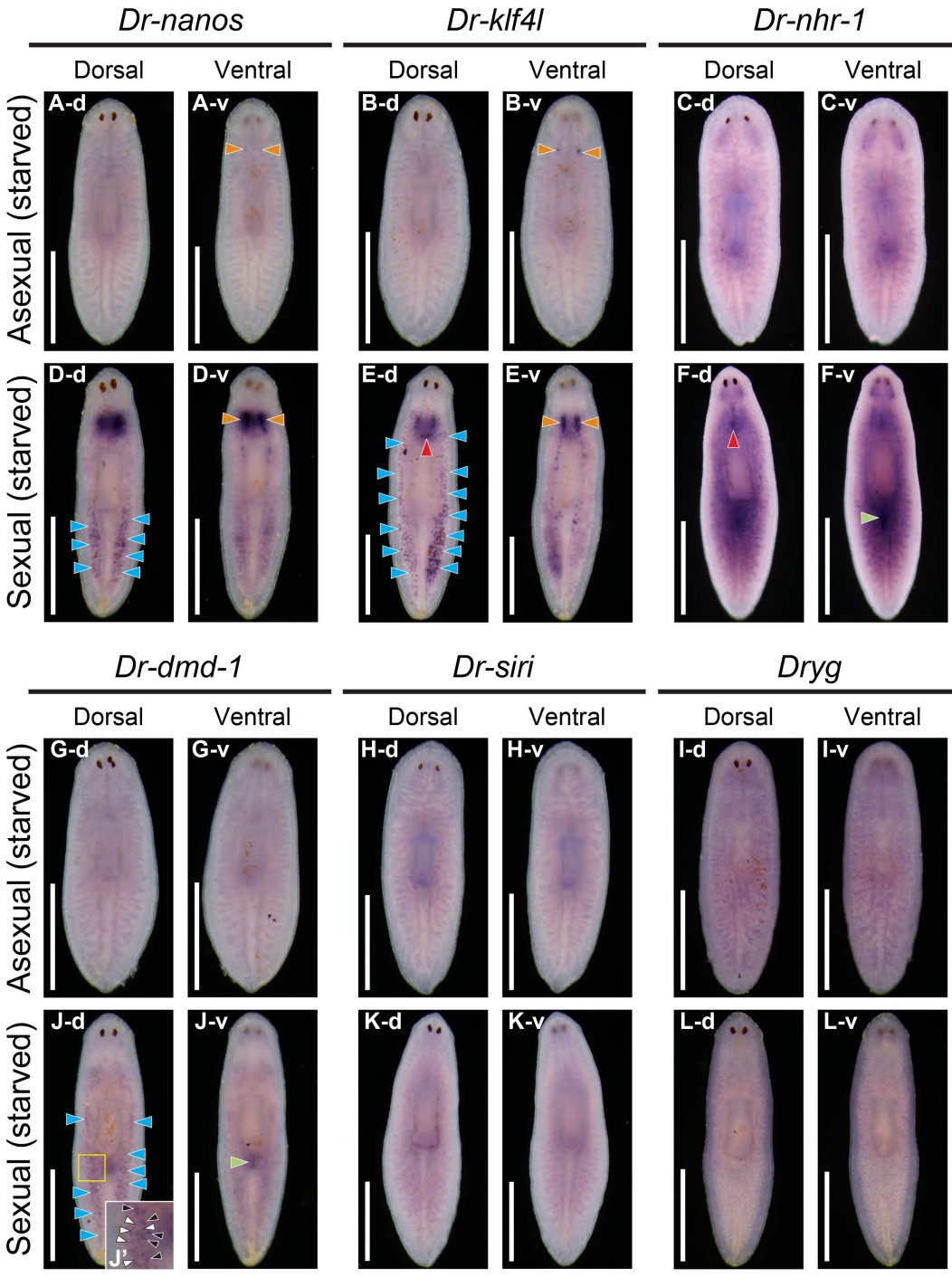

**Fig 14. Expression patterns of the essential genes for sexualization in starved sexual worms.** Starved asexual and sexual worms (3–5 mm in length) were used for samples of whole-mount *in situ* hybridization. The expression pattern was judged based on 5–8 replicates. Signals were seen as blue/purple staining. A scale bar, 1 mm. (A, D) *Dr-nanos*, (B, E) *Dr-klf4l*, (C, F) *Dr-nhr-1*, (G, J) *Dr-dmd-1*, (H, K) *Dr-siri*, (I, L) a vitellaria marker gene, *Dryg*. A pair of signals in the ovarian differentiation region, a signal in the differentiation region of a copulatory apparatus, and dot-like signals in the testicular differentiation region are indicated by orange, green, and blue arrowheads, respectively. In Fig J-d, the high magnification of the yellow box is shown in J'. Dot-like signals of *Dr-dmd-1* are indicated by black arrowheads. Moreover, a reticulate pattern of signals in the periphery of the dot-like signals was also observed (white arrowheads). Dot-like signals of *Dr-klf4l* and *Dr-nhr-1* on the dorsal midline are indicated with red arrowheads. Four of the five starved sexual worms had *Dr-klf4l* expressed in the dorsal midline, and all five starved sexual worms had *Dr-nhr-1* expressed in the dorsal midline.

On the other hand, *Dr-nanos* was not expressed in the dorsal midline of starved sexual worms in the present analysis, but we have previously confirmed its expression [66]. Although *Dr-klf4l* expression can also be observed in the dorsal midline of asexual worms (Fig 3J), no such signal was observed in the present analysis.

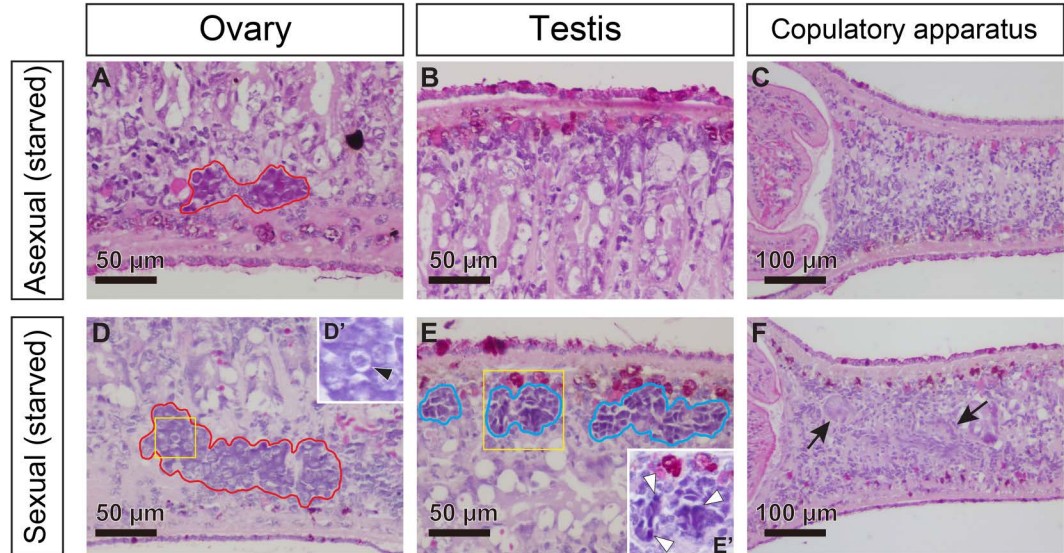

**Fig 15. Primordial testes were maintained in starved sexual worms.** Histological examination of starved asexual and sexual worms was performed. Starved worms (3–5 mm in length) were embedded, sagittally sectioned, and stained with hematoxylin and eosin (HE). The dorsal sides are at the top. (A–C) starved asexual worms (D–F) starved sexual worms. Domains bound by red and blue lines are the female germ cell masses (ovaries) and male germ cell masses (testes), respectively. Retracting copulatory organs are indicated with black arrows. (D′) high magnification of the yellow box in Fig D. A dividing cell during meiosis in the ovary is indicated by a black arrowhead, as characterized by the nuclei with clumped chromatin darkly stained with HE. (E′) high magnification of the yellow box in Fig E. Spermatogonium-like cells are indicated with white arrowheads.

planarian *S. mediterranea* testes contain somatic reproductive tissues (somatic niche cells) in which *Smed-dmd-1* and/ or *ophis* (a G protein-coupled receptor gene) are expressed [50,67]. In young juveniles and head regenerates of sexual strains (2 weeks of posterior regeneration), *Smed-dmd-1+/ ophis +* and *Smed-dmd-1+/ ophis –* cells were also observed in the presumptive differentiation region of testes. *Smed-dmd-1* and *ophis* are involved in the maintenance and differentiation of gamete stem cells (GSCs) in which a *nanos* ortholog is expressed, respectively. Therefore, the *Smed-dmd-1+/ ophis+* cells can be considered functionally mature niche cells (See Fig 8 in [67]). The fact that *Dr-dmd-1+* cells were observed in the presumptive differentiation region of testes before the appearance of the testicular primordium (Fig 8B and 8D) is consistent with the observation of *Smed-dmd-1+* cells in young juveniles and head regenerates in the sexual strain of *S. mediterranea*. Unlike *S. mediterranea*, the signals of *Dr-dmd-1* in the testes of *D. ryukyuensis* were extremely reduced after week 4 of sexualization (Figs 3I and 8). In contrast, after week 2 of sexualization, the dot-like signals of *Dr-klf4l* and *Dr-nanos* were stronger (Fig 8D), and eventually both signals of *Dr-klf4l and Dr-nanos* appeared to overlap in testes (Fig 8G). In *S. mediterranea*, *klf4l* was expressed in *nanos+* cells in testes, ovaries, and vitellaria [52]. Paraffin section FISH analysis revealed that there were *Dr-klf4l +/ Dr-nanos +* cells and *Dr-klf4l –/ Dr-nanos +* cells in the testes of *D. ryukyuensis* (Fig 12A). In the knockdown of *Dr-klf4l and Dr-nanos*, differentiation of both male and female germ cells failed in sexualization (Figs 5 and 6; [55]). Thus, we suggest that *Dr-klf4l +/ Dr-nanos +* cells in testes are GSCs in *D. ryukyuensis* similar to *S. mediterranea* [52]. Meanwhile, we found that the respective knockdowns of *Dr-dmd-1 and Dr-klf4l* did not affect each other's expression (Fig 7B and 7C). Moreover, this study revealed that *Dr-nanos* expression in the *Dr-dmd-1*

(RNAi) worms did not differ from that of control worms, while that in the *Dr-nhr-1* (RNAi) and *Dr-klf4l* (RNAi) worms was significantly reduced compared to that of the *Dr-dmd-1* (RNAi) worms (Fig 7D). These results suggest that, in the presumptive differentiation region of testes during sexual induction, *Dr-dmd-1,* expressed in the precursor cells of niche cells, and *Dr-klf4l* and *Dr-nanos,* expressed in the precursor cells of GSCs, are independently involved in testicular differentiation downstream of *Dr-nhr-1* (Fig 16).

### *Dr-siri* was identified as another novel essential gene required for sexualization

Gap junctions are involved in somatic-germ cell communication in many organisms. Studies on *C. elegans*, *D. melanogaster*, and mammals have shown that gap junctions formed by innexins and connexins are crucial in regulating somatic cell-germ cell communication [57,69–73]. In *C. elegans*, five innexins establish soma-germline gap junctions and are required for the proliferation and differentiation of germline, gametogenesis, and early embryogenesis [74–78]. Particularly, INX-14 and 21 are present in germ cell hemichannels to form gap junctions with somatic cell hemichannels formed by INX-8 and

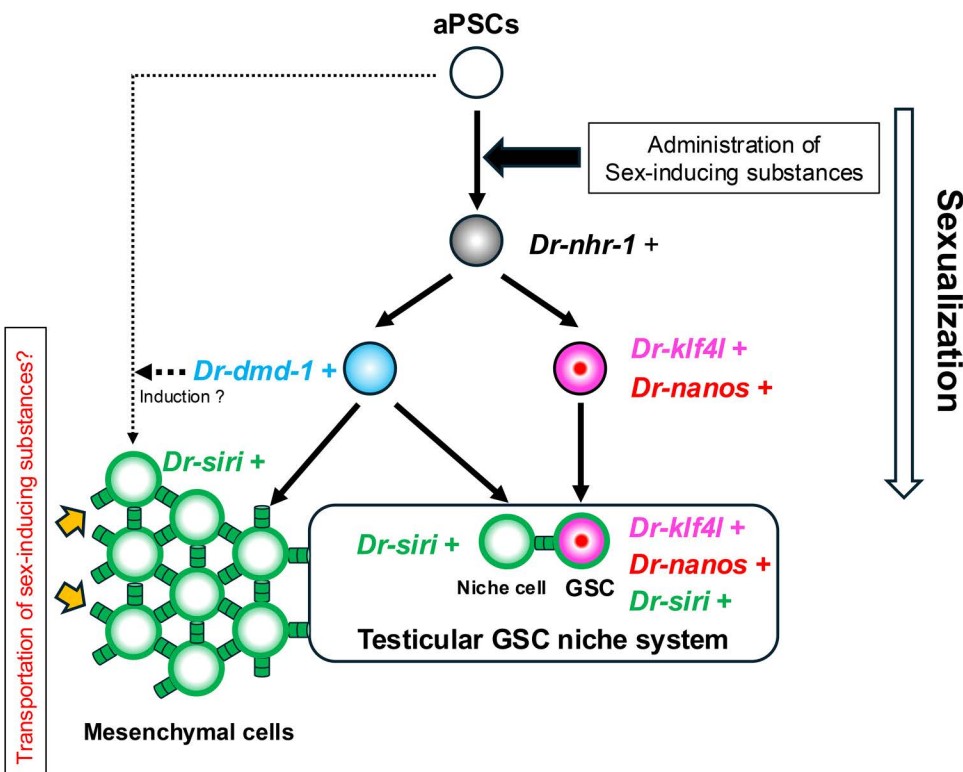

**Fig 16. Hypothetical model of sexualization in *Dugesia ryukyuensis*.** Some aPSCs in the presumptive differentiation region of testes differentiate into the cells expressing *Dr-nhr-1* by stimulation of the sex-inducing substances. Subsequently, some cells expressing *Dr-nhr-1* may differentiate to be divided into cells expressing *Dr-klf4l* or *Dr-dmd-1*. The cells expressing *Dr-klf4l* express *Dr-nanos* to precursor cells of GSCs in the presumptive differentiation region of testes during sexual induction, and the precursor cells of GSCs in which *Dr-klf4l* and *Dr-nanos* expressed may express *Dr-siri* to differentiate into mature GSCs in testes. Some of the cells expressing *Dr-dmd-1* may differentiate to precursor cells of niche cells in the presumptive differentiation region of testes during sexual induction, and the precursor cells of niche cells in which *Dr-dmd-1* expressed may disappear *Dr-dmd-1* and instead express *Dr-siri* to differentiate into mature niche cells. The possibility that mesenchymal cells expressing *Dr-siri* can be induced by *Dr-dmd-1*-expressing cells to differentiate directly from aPSCs cannot be ruled out (dotted line). However, considering that there were *Dr-dmd-1*-expressing cells that appeared in a reticulate pattern in the mesenchymal region, it is likely that the cells expressing *Dr-siri* stop expressing *Dr-dmd-1* to differentiate into a certain group of mesenchymal cells, like the differentiation of niche cells in testes. Sex-inducing substances may be transported through gap junctions containing Dr-siri protein (green-colored pipes) to work in testicular differentiation.

9. The gap junctions are required for germline proliferation [76]. In the testes of *D. melanogaster, Zpg* (Zero Population Growth, Inx4) is present in germ cell hemichannels coupled with Inx2 in neighboring somatic cells and forms gap junctions composed of two different components, innexins. Similar to the gap junctions in *C. elegans*, they are required for germ cell maintenance and regulating proliferation and differentiation of both germ and somatic cells [57,60].

The most striking finding of this study was that testicular differentiation-specific innexin *Dr-siri* was required for terminating an asexual state in switching between vegetative and sexual reproduction in planarian *D. ryukyuensis* (Figs 9–11). It is also suggested that gap junction-mediated somatic and germ-cell communication is essential for gonad differentiation in even the primitive metazoan planarians. Besides, there are several unique characteristics of *Dr-siri* of a planarian innexin. Firstly, unlike innexins, which are involved in the maintenance and differentiation of the germline stem cell niche system in *C. elegans* and *D. melanogaster* [57,60,76,77], *Dr-siri,* one of the innexin genes, appeared to be expressed in both somatic cells, including niche cells, and germ cells, including GSCs in testes (Figs 12 and 13). In this study, it was shown that *Dr-siri* was co-expressed with *Dr-klf4l* and *Dr-nanos*, which are expressed in germline cells, including GSCs in the planarian testis [52,55,63,64] (Figs 12B, 12C and 13A, 13B; yellow arrowheads). Contrary to our expectations, *Dr-dmd-1*, which was expected to be expressed in somatic niche cells, was scarcely detected in the testes of sexually mature worms (Figs 12D–E and 13C). However, paraffin section FISH analysis on *Dr-nanos* or *Dr-klf4l* and *Dr-siri* revealed that not only cells in which *Dr-nanos* or *Dr-klf4l* and *Dr-siri* co-expressed, but also cells in which *Dr-siri* was expressed alone exist in the testis (Fig 12B and 12C). We suggest that the cells expressing *Dr-siri* alone in the testes of *D. ryukyuensis* are somatic cells, including niche cells.

Secondly, planarian testes were not fully formed by the knockdown of *Dr-siri* (Fig 11). In *C. elegans* and *D. melanogaster* whose germ cells can be specified by maternally inherited determinants (preformation) [79], mutant and knockdown analyses of the relevant innexin genes cause significant effects on maintenance and differentiation of GSCs, but do not eliminate the gonads themselves [57,76,77,80,81]. As not only niche cells but also GSCs are post-embryonically differentiated and maintained from planarian aPSCs based on a conserved germline multipotency program (GMP) [82–85], mature testes would not be maintained without the gap junction-mediated somatic and germ cell communication by Dr-siri protein.

Thirdly, as far as we know, innexins specifically involved in testicular differentiation, such as *Dr-siri*, have not been reported. *Smed-dmd-1* was also identified as a molecule involved in male identity in even hermaphrodites such as planaria [50]. The phenotype in the knockdown of *Dr-siri* was almost identical to that of *Dr-dmd-1* (Figs 5–6, and 11). Thus, we inferred that *Dr-siri* is also involved in male identity. Orthologs of *Dr-siri* may be widely implicated in determining males in other animals.

## Gap junction involved in the Dr-siri protein may transport sex-inducing substances for testicular differentiation

In WISH analysis, before the point of no return, dot-like signals of *Dr-siri* were recognized in a reticulate pattern in the presumptive differentiation region of testes, like those of *Dr-dmd-1*, *Dr-klf4l*, and *Dr-nanos* (Figs 8B and 10D, black arrowheads). After the point of no return, the strong signals of *Dr-siri* were observed in the cell mass-like structures in the testicular differentiation region (Fig 10D, black arrowheads). Paraffin section and whole-mount FISH analyses of sexual worms revealed that *Dr-klf4l* and *Dr-nanos* were co-expressed, and *Dr-siri* was co-expressed with *Dr-klf4l* or *Dr-nanos* in the testis (Figs 12A–C and 13A, B; yellow arrowheads). We expect that the precursor cells of GSCs in which *Dr-klf4l* and *Dr-nanos* expressed may express *Dr-siri* to differentiate into mature GSCs in testes (Fig 16). Furthermore, we found cells expressing *Dr-siri* alone in the testis of sexually mature worms (Fig 12B and 12C; white arrowheads), though cells expressing *Dr-dmd-1* were hardly recognized (Figs 12D–E and 13C). We expect that the precursor cells of niche cells in which *Dr-dmd-1* expressed may disappear *Dr-dmd-1* and instead express *Dr-siri* to differentiate into mature niche cells (Fig 16).

Moreover, we found that *Dr-siri* alone is expressed in the mesenchymal cells around the testes (Figs 13; white arrowheads and S10). A reticulate pattern of *Dr-siri* signals recognized in the periphery of testes (Fig 10D, white arrowheads) was very similar to the expression pattern of *Dr-dmd-1* in weeks 2–3 of sexualization (Fig 8D and 8E, white arrowheads). Although the possibility that mesenchymal cells expressing only *Dr-siri* can be induced by *Dr-dmd-1*-expressing cells to

differentiate directly from aPSCs cannot be ruled out, some cells expressing *Dr-siri* may stop expressing *Dr-dmd-1* to differentiate into a certain group of mesenchymal cells (Fig 16), because *Dr-dmd-1* was expressed in a reticulate pattern in the mesenchymal region in addition to the dot-like *Dr-dmd-1*-expressing cells that appeared in the niche region.

We found that there are at least 46 innexin genes in the transcriptome catalogs of *D. ryukyuensis* (S7 Table). The number of these innexin genes is far greater than that of *C. elegans* and *D. melanogaster*. As planarians do not have a vascular system, they may have evolved a unique form of low molecular compound transport that utilizes gap junctions. *Dr-siri* was clearly expressed widely in mesenchymal regions other than the testes (Figs 10B, 13 and S10). The sex-inducing substances, whose molecular weight is estimated to be around 500 [46], may be transported throughout the body through gap junctions by certain innexins to work in reproductive organ differentiation and maturation.

During starvation of sexual worms, somatic organs such as the pharynx and eyes remained intact, whereas reproductive organs became immature (Figs 14 and 15). The starved sexual worms again develop mature reproductive organs even when fed beef or chicken liver, a common food for planarians that has no sex-inducing activity [66]. In starved sexual worms, *Dr-dmd-1*, *Dr-klf4l*, and *Dr-nanos*-positive cells were found in the testicular primordia, but little *Dr-siri* expression was observed (Fig 14). The re-expression of *Dr-dmd-1* in starved sexual worms is noteworthy because *Dr-dmd-1* was down-regulated once it reached a mature sexual state (Figs 3I, 8, 12 and 13). As *Dr-siri* is expected to be expressed downstream of *Dr-dmd-1*and *Dr-klf4l* in the somatic cells in mesenchymal space (including the niche cells) and the GSCs, respectively (Fig 10F), we can say that *Dr-siri* expression is in "standby mode" in starved sexual worms. The reason why the differentiated state of the reproductive organs is not maintained in vain during starvation may be that gap junctions, mediated by certain innexins, including the Dr-siri protein, are temporarily lost, and sex-inducing substances and other essential substances are no longer transported throughout the body. Innexin genes other than *Dr-siri* involved in reproductive organ differentiation would be contained in 46 innexin homologs in *D. ryukyuensis.*

The starved sexual worms that maintain their acquired sex possessed testicular primordia but not mature vitellaria (Figs 14 and 15). Previously, we indicated that sex-inducing substances are present in vitellaria, though it is not yet known where the sex-inducing substances are synthesized [86]. *Dr-nhr-1, Dr-dmd-1*, *Dr-klf4l*, and *Dr-siri* (RNAi) worms could not differentiate testes (Figs 5–6, and 11) and could not establish and maintain a sexual state (Figs 5B and 11A), suggesting that testicular differentiation may be associated with the termination of an asexual state and the maintenance of acquired sex, and that vitellaria and the sex-inducing substances pooled in vitellaria do not appear to be related to the maintenance of acquired sex. However, it is not clear whether testes are involved in the production of sex-inducing substances. Considering all of the above together, gap junctions containing the Dr-siri protein may establish a testicular niche system by transporting experimentally administered and endogenous sex-inducing substances and/or other required substances for testicular maturation from outside of the testes (Fig 16). For reproductive organs other than the testes, the maturation may also be regulated by a similar system of gap junctions constituted by other innexins, because differentiation of ovaries and vitellaria were not affected in the RNAi treatment of *Dr-siri* with sexualization (Fig 11). However, the gap junctions containing the Dr-siri protein are also likely to be associated with the differentiation of the copulatory apparatus as well as testes (Fig 11).

The changes that dramatically reduce the expression of the *dmd* homolog of *D. ryukyuensis* in dorsal mesenchymal space (including testicular niche cells) as test worms mature have not occurred in that of *S. mediterranea* [50]. Oddly enough, the homologs in *D. ryukyuensis* for *npy-8* (neuro peptide Y-8) and *npyr-1* (neuro peptide Y receptor-1), which are involved in testis differentiation in *S. mediterranea* [87], do not differ significantly in expression between the asexual and sexual worms (S11 Fig). We found several *npy* homologs other than *npy-8* that are more strongly expressed in sexual than asexual worms. However, these *npy* homologs were expressed in the copulatory apparatus, not in testes, and knockdown of the *npy* homologs did not result in a phenotype, suggesting that the planarian testicular niche system contains not only common mechanisms that are conserved across species but also unique mechanisms. We will examine whether *Dr-siri* orthologs involved in testicular differentiation and sexualization are present in *S. mediterranea* and a closely related species, *D. japonica*.

Meanwhile, the sex-inducing substances are found not only in planaria but also in distantly related parasitic flatworms, suggesting that the molecular mechanisms associated with sex-inducing substances are also conserved. Thus, the testicular stem cell niche system supported by *Dr-siri* orthologs may be involved in the reproductive switching of the parasitic flatworms. In the future, we plan to examine whether the reproductive switching of *Schistosoma mansoni* and *Fasciola* sp., in which sexualizing activity against *D. ryukyuensis* was recognized [44], is regulated by a similar mechanism to that used in the planarians.

## Materials and methods

### Organisms

An exclusively asexual strain of the planarian *D. ryukyuensis*, the OH strain, was established in 1986 by Dr. S. Ishida at Hirosaki University (Hirosaki, Japan). The OH strain was maintained at 20 °C in autoclaved tap water and fed chicken liver once a week. Populations of *B. brunnea* were collected at Liver Tsuchibuchi in Hirosaki City, Japan. They were frozen in liquid nitrogen and then stored at -80 °C as a source of sex-inducing substances. Worms of the OH strain were sexualized by feeding sexually mature worms of *B. brunnea* or an extract from *B. brunnea*, the fraction M0 + M10, as described previously [41,46].

### Preparation of the fraction M0 + M10

Approximately 2 g wet weight of sexually mature worms of *B. brunnea* was used. It was homogenized in 120 mL PBS (34 mM NaCl, 0.68 mM KCl, 2.5 mM $Na_2HPO_4$, and 0.45 mM $KH_2PO_4$; pH 7.4). The homogenate was centrifuged at 16,000 × $g$ for 30 min at 4 °C. The supernatant was filtered using a 0.22 μm MillexGV filter (Millipore, Carrigtwohill, Cork, Ireland) and then centrifuged at 120,000 × $g$ for 30 min at 4 °C. It was subsequently freeze-dried and dissolved in water, yielding 18 mL of extract. The extract was further loaded onto a Sep-Pak Light t$C_{18}$ Cartridge (Waters, Milford, MA, USA). The sex-inducing substances were sequentially eluted using 0% and 10% aqueous methanol to create 48 mL of Fr. M0 and Fr. M10, respectively. The obtained fractions were mixed and freeze-dried, and the resulting powders were used in feeding bioassays [46,88].

### RNA-seq

In this study, we performed two distinct projects of RNA-seq: RNA-seq focusing on the point of no return and RNA-seq of the knocked-down test worms of three essential genes required for transgressing the point of no return.

### Identification of 12 genes including *Dr-nhr-1*, *Dr-dmd-1*, and *Dr-klf4l*

In the RNA-seq focusing on the point of no return, we prepared food containing the Fr. M0 + M10 derived from approximately 2 g wet weight of *B. brunnea.* Approximately ten OH worms were fed daily with a piece of the food divided into 45 pieces for 3 d, 1 week, 2 weeks, and 3 weeks. The analysis was performed in triplicate to develop an RNA-seq library (S1 Table). After "regeneration test after sexualization" (S1 Fig), total RNA extracted using Sepasol RNA I Super G (Nacalai Tesque, Kyoto, Japan) following the manufacturer's instructions was prepared from control worms, day 3-worms, week 1-worms, week 2 before-worms, week 2 after-worms, and week 3-worms. Total RNA was treated with TURBO DNase using the TURBO DNA-free kit (Thermo Fisher Scientific, Waltham, MA, USA) and was purified using the RNeasy micro kit (QIAGEN, Hilden, Germany), following the manufacturer's recommendations. RNA integrity was validated using an Agilent 2100 Bioanalyzer (Agilent Technologies, Santa Clara, CA, USA). The cDNA libraries were prepared using the TruSeq RNA sample preparation kit v2 (Illumina, San Diego, CA, USA) following the manufacturer's instructions ("Low Sample Protocol"). Briefly, mRNA was purified from 0.5 μg of total RNA using oligo-dT magnetic beads and chemically fragmented. The cDNA was synthesized using SuperScript II reverse

transcriptase (Invitrogen, Carlsbad, CA, USA) and random primers. The resultant cDNA was purified using AMPure XP beads (Beckman Coulter, Brea, CA, USA). The cDNA was then subjected to end-repair processing, and the 3′-ends were adenylated and ligated with paired-end adaptors (Illumina). The cDNA fragments were amplified using adaptor-specific primers (Illumina). The enriched cDNA libraries were validated using an Agilent 2100 Bioanalyzer (Agilent Technologies). Multiplex sequencing of paired-end reads was performed using an equimolar mixture of the final cDNA libraries and an Illumina HiSeq. 2000 system (Illumina).

RNA-seq data from day 3-worms, week 1-worms, week 2 before-worms, week 2 after-worms, and week 3-worms were used. Genes satisfying the following three conditions were narrowed down as DEGs: (1) being upregulated before the point of no return, (2) having a CDS region, and (3) having significantly different expression levels between asexual and sexual worms. The analysis procedure is to map the reads of each sample to the reference sequence [47] obtained via *de novo* assembly using Bowtie2 (version 2.2.6) [89] and to infer the number of reads for each contig using eXpress (version 1.5.1) [90]. Based on the Reads Count values, edgeR (version 3.12.0) [91,92] was used to calculate the DEGs between week 2 before-worms and control worms. The CDS data [47] was imported to narrow down genes containing CDSs. FPKM of sexual/ (FPKM of asexual + 0.01) was further calculated using the transcriptome catalogs of *D. ryukyuensis* [47], and genes with FPKM of sexual/ (FPKM of asexual + 0.01) >10 were narrowed down from the DEGs, and 12 genes were selected.

### Identification of 27 genes, including *Dr-siri*

In the RNA-seq of the knocked-down test worms of three essential genes required for transgressing the point of no return, total RNA was extracted from control (*egfp* (RNAi)), *Dr-nhr-1* (RNAi), *Dr-dmd-1* (RNAi), and *Dr-klf4l* (RNAi) worms (S4 Table), using Sepasol RNA I Super G (Nacalai Tesque, Kyoto, Japan) following the manufacturer's instructions. Non-targeted RNA-seq was conducted according to the Lasy-Seq ver. 1.1 protocol (https://sites.google.com/view/lasy-seq/). Briefly, 180 ng of total RNA was reverse transcribed using a reverse transcription (RT) primer with an index and SuperScript IV reverse transcriptase (Thermo Fisher Scientific). Thereafter, all RT mixtures were pooled and purified using an equal volume of AMpure XP beads (Beckman Coulter, Brea, CA, USA) according to the manufacturer's instructions. Second-strand synthesis was conducted with the pooled samples using RNaseH (5 U/µL; Enzymatics, Beverly, MA, USA) and DNA polymerase I (10 U/µL; Enzymatics). To avoid the carryover of large amounts of rRNAs, the mixture was subjected to RNase treatment using RNase T1 (Thermo Fisher Scientific). Subsequently, the samples were purified using 0.8 × volume of AMpure XP beads. Fragmentation, end-repair, and A-tailing were conducted using 5 × WGS Fragmentation Mix (Enzymatics). The Adapter for Lasy-Seq was ligated using 5 × Ligation Mix (Enzymatics), and the adapter-ligated DNA was purified twice with 0.8 × volume of AMpure XP beads. After optimizing the PCR cycles for library amplification by qPCR using EvaGreen, 20× in water (Biotium, Fremont, CA, USA) and the QuantStudio5 Real-Time PCR System (Applied Biosystems, Waltham, MA, USA), the library was amplified using KAPA HiFi HotStart ReadyMix (KAPA BIOSYS-TEMS, Wilmington, MA, USA) on the ProFlex PCR System (Applied Biosystems). The amplified library was purified with an equal volume of AMpure XP beads. A total of 1 µL of the library was subjected to electrophoresis using Bioanalyzer 2100 with the Agilent High Sensitivity DNA kit (Agilent Technologies) to assess quality. Subsequently, sequencing of 150-bp paired-end reads was performed using HiSeq X Ten (Illumina).

The reads are trimmed using fastp [93,94]. After trimming, they were mapped to transcriptome data using BWA mem [95–97]. The resulting SAM files were converted to BAM format and sorted with samtools [96]. Finally, the read count data was quantified using Salmon [98] to obtain read count data. We then used edgeR [91,92] for each knocked-down worm to determine significant differences (likelihood ratio test, FDR < 0.05), narrowing down the genes commonly upregulated in the 3 groups and the genes commonly downregulated in the 3 groups to 4 upregulated genes and 792 downregulated genes. A total of 27 genes, including *Dr-siri,* were obtained from the common genes between 792 downregulated genes and upregulated genes in week 3 of sexualization.

## Whole-mount *in situ* hybridization

To obtain partial nucleotide sequences used for the design of *in situ* hybridization probes, the target genes were cloned, as previously described [99,100]. The primers used in the current study are given in S10 Table. Digoxigenin (DIG)-labeled anti-sense RNA probes, were synthesized *in vitro* using DIG-11-UTP (Roche, Mannheim, Germany) and the MEGA script T7 (or SP6) kit (Thermo Fisher Scientific). *In situ* hybridization of whole-mount worms was performed per our previously published methods [101]. DIG-labeled probes were detected using alkaline phosphatase (AP)-conjugated anti-DIG antibodies (Roche, Mannheim, Germany) and stained blue with a NBT/BCIP solution [170 μg/mL nitro-blue tetrazolium chloride (NBT) (Roche) and 175 μg/mL 5-bromo-4-chloro-3-indolylphosphate, toluidine-salt (BCIP) (Roche)]. Specimens were examined, and images were taken using a digital microscope setup with an Olympus SZX10 microscope (Olympus Corporation, Tokyo, Japan) and a CMOS Camera Tablet PCset, CP605100B tablet (BioTools Inc., Gunma, Japan).

## Reverse-transcription quantitative PCR (RT-qPCR)

Total RNA was extracted from individual worms and treated with DNase I, as described in the RNA-seq section. Approximately 0.5 μg of total RNA was used to prepare cDNA using the ReverTra Ace kit (Toyobo, Tokyo, Japan). Approximately 200 ng of total RNA in each sample was used to prepare cDNA using a ReverTra Ace kit (Toyobo, Tokyo, Japan). Each cDNA sample was dissolved in 20 μL of water. RT-qPCR was performed using an AriaMx Real-Time PCR System (Agilent Technologies) according to our previously published methods [47]. Each reaction mixture (25 μL) contained 12.5 μL of the KAPA SYBR Fast qPCR Master Mix kit (KAPA Biosystems, Wilmington, MA, USA), 10 μM gene-specific primers, and 1.0 μL of cDNA. The elongation factor, 1 alpha homolog (*Dr-ef1a*) [48] or glyceraldehyde-3-phosphate dehydrogenase (*Dr-gapdh*) [53] was used as an internal control. S10 Table summarizes the gene-specific primers.

Differences in the obtained threshold cycle (Ct) values between samples were calculated using the ΔΔCt method. Briefly, ΔCt [where ΔCt = Ct(target gene) - Ct(internal control)] was calculated for each sample (e.g., asexual and sexual), and then ΔΔCt [where ΔΔCt = ΔCt(sample) - the average of ΔCt(calibrator)] was calculated. Calibrators were the asexual worms or the control worms. Statistical tests were performed on the ΔΔCt values. Relative expression was calculated as 2-ΔΔCt.

## Statistical analysis

Statistical tests were performed using R v3.2.2 [102]. When gene amplification was not detected, which was often the case for sexual DEGs tested in asexual worms, expression was treated as not available (NA) in the calculations. The Shapiro–Wilk test was used to validate the normal distributions of obtained data, and the F-test or Berlett's test was used to validate equality of variances; then, Student's *t*-test was used to compare gene expression levels between two groups (e.g., asexual and sexual samples, or control and knockdown samples). In a few cases, Welch's t-test was used because of unequal variances between the samples. To compare gene expression levels among three or more groups, Tukey's honestly significant difference (HSD) test was used.

## RNAi gene silencing

Double-stranded RNAs (dsRNAs) of the candidates for essential genes required for transgressing the point of no return were *in vitro*-synthesized using a MEGAscript High Yield Transcription kit (Ambion, Austin, TX, USA). *Enhanced green fluorescent protein (egfp)* dsRNA was synthesized as a control dsRNA. S10 Table summarizes the primers for dsRNAs. In the knockdown experiment (Fig 5A), we set the dsRNA concentration at 100 ng/day/worm. In the pretreatment of knockdown with sexualization, dsRNA was mixed with chicken liver homogenate (approximately 10 μL/worm) and freeze-dried for use as food for asexual worms of the OH strain (test worms). During the pretreatment of knockdown with sexualization,

test worms were fed a piece of the food five times. To silence the candidate genes during sexualization, the test worms were fed dsRNA mixed with *B. brunnea* homogenate (approximately 10 μL/day/worm) daily for 5 weeks.

## Histology

Worms were relaxed in cold 2% (v/v) HCl in 5/8 Holtfreter's solution [103] for 5 min and then fixed in 4% paraformaldehyde and 5% methanol in 5/8 Holtfreter's solution (pH 7.4) for 3 h at 20 °C. The fixed specimens were dehydrated via an ethanol series, cleared in xylene, and embedded in a Paraplast Plus embedding medium (Sigma-Aldrich Co., St. Louis, MO, USA). The embedded specimens were cut into 4 μm thick sections and stained with Mayer's Hematoxylin Solution (Wako, Osaka, Japan) and Eosin Y (yellowish), Certistain for microscopy (Merck, Darmstadt, Germany). Specimens were examined, and images were taken using a digital microscopy setup involving a Nikon Eclipse E800 microscope (Nikon, Tokyo, Japan) and a CMOS Camera Tablet PCset, CP605100B tablet (BioTools Inc.).

## Paraffin section fluorescence *in situ* hybridization (FISH)

Samples prepared for histology were sectioned to a size of 10 μm in the sagittal direction. After hydration, the samples were incubated for 15 min in 500 mL of 0.1 M triethanolamine / 0.25% HCl, during which 1.25 mL of acetic anhydride was added dropwise to the solution. To obtain partial nucleotide sequences used for the design of FISH probes, the target genes were cloned. The primers used in the current study are given in S10 Table. Digoxigenin (DIG)- or fluorescein (FITC)-labeled antisense RNA probes were synthesized *in vitro* using DIG-11-UTP or FLU-12-UTP (Roche, Mannheim, Germany) and the MEGA script T7 (or SP6) kit (Thermo Fisher Scientific). DIG-labeled probes were detected using POD-conjugated anti-DIG antibodies, and FITC-labeled probes were detected using POD-conjugated anti-FITC antibodies (Roche, Mannheim, Germany). Hybridization was conducted using digoxygenin (DIG)- or fluorescein (FITC)-labeled probes, as previously described [104]. Excess probes were washed out, and the samples were incubated with 1% blocking reagent (Roche, Mannheim, Germany). For fluorescent staining, samples were incubated with 1/100 diluted anti-DIG-peroxidase (POD) (Roche, Mannheim, Germany) or 1/10 diluted anti-FITC-POD (Roche, Mannheim, Germany) antibody overnight. After washing with borate buffer (100 mM borate (pH 8.5), 0.1% Tween-20), detection was conducted in tyramide signal amplification reaction solution ((Alexa Fluor 488, or Alexa Fluor 594, Invitrogen, Carlsbad, CA, USA), 0.003% $H_2O_2$, 2% dextran sulfate sodium salt, and 0.3 mg/ml 4-iodo-phenol in borate buffer) [105,106]. After the tyramide signal amplification reaction, samples were incubated with 5 μg/mL Hoechst33342 solution at 4 °C for o/n. Finally, they were mounted with Fluorokeeper (Nacalai Tesque, Kyoto, Japan). Specimens were examined, and images were taken using a digital microscopy setup with an Olympus FV3000 (Olympus Corporation, Tokyo, Japan).

## Whole-mount FISH

Until the pre-hybridization step, FISH of whole-mount worms was performed according to our previously published methods [101]. To optimize the fluorescence staining, we washed samples with borate buffer (100 mM borate [pH 8.5], 0.1% Tween-20), and conducted tyramide signal amplification in TSA reaction solution (tyramide reagent, 0.003% $H_2O_2$, 2% dextran sulfate sodium salt, and 0.3 mg/mL 4-iodo-phenol in borate buffer) [105,106]. After the TSA reaction, samples were incubated with Hoechst 33342. After transparency treatment (glycerol, 1 M Tris-HCl [pH 7.5], 0.5 M EDTA, DW, 20 h), we washed samples with copper sulfate solution, then performed transparency treatment again. Specimens were sealed with Fluorokeeper (Nacalai Tesque, Kyoto, Japan). Specimens were examined, and images were taken using a digital microscopy setup with an Olympus FV3000 (Olympus Corporation, Tokyo, Japan).

## Supporting information

**S1 Fig. Experimental scheme of "regeneration test" for the point of no return.** In this study, the samples for RNA-seq library and knockdown experiments were checked using the "regeneration test after sexualization." Step 1: To sexualize, the test worms (the asexual OH worms) were daily fed with sexually mature worms of *Bdellocephala brunnea*

or the fraction M0 + M10 that contains the sex-inducing substances from *B. brunnea* during each experimental period. Step 2: After the feeding assay, the test worms were transversely cut into two pieces at the prepharyngeal level. Step 3: Immediately after cutting, total RNAs were extracted from the head fragments of half of the test worms and the tail fragments of the other half. Step 4: Fragments from which RNA was not extracted were allowed to regenerate for approximately 1 month by being fed with beef liver, which does not have sex-inducing activity. Step 5: The reproductive modes of the regenerates were examined. If the test worms have not transgressed the point of no return (e.g., if they are stage 2-worms), both the head- and tail-derived fragments become asexual after regeneration. In contrast, if they have transgressed the point of no return (e.g., if they are stage 3-worms), both the head- and tail-derived fragments become sexual after regeneration. Thus, observation of the reproductive mode of the regenerated fragment can retrospectively determine whether the worms had exceeded or not the point of no return at the time of cutting, for example, even if the morphological features of stage 3 were not developed yet. Step 6: When the reproductive mode of the regenerates was the same, total RNAs extracted from the head and tail fragments of the corresponding other half were mixed. The mixed RNA samples, which were considered as RNAs from one individual, were used for developing the RNA-seq library and RT-qPCR analysis.

(TIF)

**S2 Fig. *Dr-dmd-1* (TR29049|c0_g1_i1) is the homolog of *dmd-1* in *S. mediterranea.*** In this study, we named TR29049|c0_g1_i1 *Dr-dmd-1*. (A) Five genes annotated as double sex and mab3 transcription factor were contained in the transcriptome catalogues of *D. ryukyuensis* (Sekii et al., 2019). Top hits obtained using Protein BLAST search (https://blast.ncbi.nlm.nih.gov/Blast.cgi) of these DM domain genes as the query are shown. (B) Phylogenic trees for the DM domain genes in *D. ryukyuensis* using either the maximum likelihood method (upper) or the neighbor-joining method (lower). The percentage of trees in which associated taxa clustered together in the bootstrap (1000 replicates) is shown next to the branches. Scale bars: upper, substitutions per site; lower, evolutionary distance. The maximum likelihood and the neighbor-joining trees were constructed using MEGA version 12.0.11. Accession numbers: *Dmd-1* splice form 1, AGL61623; *Dmd-1* splice form 2, AGL61624; *Dmd-1* splice form 3, AGL61625; *Dmd-1* splice form 4, AGL61626; *Dmd-2* splice form 1, AGL61627; *Dmd-2* splice form 2, AGL61628; *Dmd-3*, AGL61629; *Dmd-4*, AGL61630; *Doublesex*, NP_001262353; *Mab 3*, NP_001022464. (C) The FPKM values in asexual and sexual worms of the four DM domain genes other than TR29049|c0_g1_i1. Bar graphs were plotted based on fragments per kilobase of exon per million reads mapped (FPKM) value in the RNA-seq data of *D. ryukyuensis* (Sekii et al., 2019). Two races of sexual worms of *D. ryukyuensis* occur (Kobayashi et al., 2012). Ac-Sexual: Acquired sexual worms can switch to an asexual state; In-Sexual: Innate sexual worms cannot switch to an asexual state.

(TIF)

**S3 Fig. Regeneration test after feeding minced worms of *B. brunnea.*** The test worms were fed with minced worms of *B. brunnea* for 3 d, 1 week, 2 weeks, or 3 weeks. After feeding with minced worms of *B. brunnea*, the test worm was subjected to a regeneration test shown in S1 Fig. If the fragments become sexual after the regeneration test, it can be determined retrospectively that the worm had exceeded the point of no return at the time of cutting (i.e., after feeding the worm with minced worms of *B. brunnea* for each duration). Similarly, if the fragments become asexual, it was determined that the worm had not exceeded the point of no return.

(TIF)

**S4 Fig. *Dr-nhr-1* (TR20737|c0_g1_i1) is the homolog of *nhr-1* in *S. mediterranea.*** In the current study, we named TR20737|c0_g1_i1 *Dr-nhr-1*. (A) We performed a local BLASTX search for the transcriptome catalogues of *D. ryukyuensis* by using 23 *nhr* genes of *S. mediterranea* as the query with an e-value cut-off of $10^{-30}$, resulting in ten homologs. The description of the DNA-binding domain similarity to human was reproduced from Table S1 of Tharp et al. (2014). (B) The FPKM values in asexual and sexual worms of the nine *nhr* genes other than TR20737|c0_g1_i1. Bar graphs were

plotted based on FPKM value in the RNA-seq data of *D. ryukyuensis* (Sekii et al., 2019). Two races of sexual worms of *D. ryukyuensis* occur (Kobayashi et al., 2012). Ac-Sexual: Acquired sexual worms can switch to an asexual state; In-Sexual: Innate sexual worms cannot switch to an asexual state. (C) Phylogenetic trees for the planarian *nhr* genes and the *nhr* genes in other animals using either the maximum likelihood method (left) or the neighbor-joining method (right). The percentage of trees in which associated taxa clustered together in the bootstrap (1000 replicates) is shown next to the branches. Scale bars: left, substitutions per site; right, evolutionary distance. The maximum likelihood and the neighbor-joining trees were constructed using MEGA version 12.0.11. The sequences of 23 *nhr* genes in *S. mediterranea* in the supplementary data of Tharp et al. (2014) were used. Accession numbers: estrogen-related receptor alpha [*Mus musculus*], AAB51250; estrogen-related receptor, isoform B [*Drosophila melanogaster*], NP_648183; nuclear hormone receptor family member nhr-6 [*Caenorhabditis elegans*], NP_497731; nuclear receptor 2DBD-gamma [*Schistosoma mansoni*], AAW88550; steroid hormone receptor ERR1 isoform 2 [*Homo sapiens*], NP_001269380.
(TIF)

**S5 Fig. *Dr-klf4l* (TR16868|c0_g1_i1) is the homolog of *klf4l* in *S. mediterranea*.** In this study, we named TR16868|c0_g1_i1 *Dr-klf4l*. (A) We obtained TR16868|c0_g1_i1 as the top hit through a Protein BLAST search of the deduced amino acid sequence of *klf4l* in *S. mediterranea* as the query against the transcriptome catalogues of *D. ryukyuensis* (Sekii et al., 2019). The amino acid sequence homology between *Dr-klf4l* (TR16868|c0_g1_i1) and *klf4l* in *S. mediterranea* was analyzed using the GENETYX-MAC version 13.0.14 (GENETYX, Tokyo, Japan). The amino acid sequence of *Dr-klf4l* is 54.7% (265 amino acids) identical to that of *klf4l* in *S. mediterranea*. The matching amino acids are shaded in black. (B) Phylogenetic trees for the planarian *klfl* genes and the krüppel-like factor (KLF) family in human and mouse using either the maximum likelihood method (left) or the neighbor-joining method (right). The percentage of trees in which associated taxa clustered together in the bootstrap (1000 replicates) is shown next to the branches. Scale bars: left, substitutions per site; right, evolutionary distance. The maximum likelihood and the neighbor-joining trees were constructed using MEGA version 12.0.11. Accession numbers: KLF1_human, NP_006554; KLF1_mouse, NP_034765; KLF2_human, NP_057354; KLF2_mouse, NP_032478; KLF3_mouse, NP_032479; KLF4_human, NP_001300981; KLF4_mouse, NP_034767; KLF5_human, NP_001721; KLF5_mouse, NP_033899; KLF6_human, NP_001291; KLF6_mouse, NP_035933; KLF7_human, NP_003700; KLF8_human, NP_001311031; KLF8_mouse, NP_001344112; KLF9_human, NP_001197; KLF10_human, NP_001027453; KLF10_mouse, NP_001276400; KLF11_human, NP_001171187; KLF11_mouse, NP_848134; KLF12_human, NP_001387075; KLF12_ mouse, NP_001398638; KLF13_human, NP_057079; KLF14_human, NP_619638; KLF15_human, NP_054798; KLF15_mouse, NP_001342597; KLF16_human, NP_114124; KLF16_mouse, NP_510962; KLF17_human, NP_775755; KLF17_mouse, NP_083692; KLF18_human, NP_001345367.
(TIF)

**S6 Fig. Functional analysis of candidate essential genes in sexualization.** No phenotypes were observed in the knock-down worms of the candidate genes other than *Dr-nhr-1, Dr-dmd-1*, *Dr-klf4l,* and TR29311|c0_g1_i1. (A) TR18170|c0_g1_i1 and TR18958|c0_g1_i1. (B) TR80154|c0_g1_i1 and TR36563|c0_g1_i1. (C) TR45646|c0_g1_i2. (D) TR4477|c0_g1_i1. The worms were evaluated through external observation, and the results are shown in a donut chart with four distinctions (Stage 0 [asexual], Stages 1–2, Stages 3–4, and Stages 5–6). The number of test worms after RNAi treatment with sexualization is shown in the center of the doughnut chart. The doughnut chart displays the number of worms at each of the four stages of sexualization in its circular sections. Live ventral images of the most sexually mature test worm are presented. Blue arrowheads highlight a pair of ovaries, while green and pink arrowheads point out the copulatory apparatus and genital pore, respectively. A scale bar, 1 mm. In the knocked-down worms, RNAi efficiency was examined using RT-qPCR. The RT-qPCR data are shown relative to the expression level in the control worm, and log2 (relative expression) on the vertical axis indicates -ΔΔCt. Each circle indicates a control or knocked-down worm. Eight

replicates were used, but data were handled as NA (not available) if the expression was in the case of outliers. The bars in the plots indicate the averages of -ΔΔCt. Asterisks indicate significant differences between the control and knocked-down worms (Student's *t*-test: ***P < 0.001). Note that the PCR datum for TR18958|c0_g1_i1 was not available because appropriate primers could not be designed.
(TIF)

**S7 Fig. Description of marker genes for a copulatory apparatus (TR44991|c0_g1_i2) and the testis (TR34243|c0_g1_i1).** (A) TR44991|c0_g1_i2 was annotated as Cathepsin L. (B) TR34243|c0_g1_i1 was a homolog of C3H-zinc finger-containing protein 1 in *S. mediterranea* (Wang et al., 2010). (A–B) Expressions in asexual and sexual worms were examined using RT-qPCR analysis. The RT-qPCR data are shown relative to the expression level in the asexual worm, and log2 (relative expression) on the vertical axis indicates -ΔΔCt. Each circle indicates an asexual or a sexual worm. Eight replicates for TR44991|c0_g1_i2 expression and five replicates for TR34243|c0_g1_i1 expression were used, but data were handled as NA (not available) if the expression was too low to be detected. The bars in the plots indicate the averages of -ΔΔCt. Asterisks indicate significant differences between the asexual and sexual worms (Student's *t*-test: ***P < 0.001). Changes in expression level during sexualization were also examined using RT-qPCR analysis. C, control worms (asexual OH worms); D3, day 3-worms; W1, week 1-worms; W2b, week 2 before-worms; W2a, week 2 after-worms; W3, week 3-worms. The RT-qPCR data for each candidate essential gene are shown relative to the expression level in the control worm, and log2 (relative expression) on the vertical axis indicates -ΔΔCt. Each circle indicates an individual worm in the control or minced *B. brunnea*-fed groups. Six replicates were used, but data were handled as NA (not available) if the expression was too low to be detected. The bars in the plots indicate the averages of -ΔΔCt. Asterisks indicate significant differences compared with control worms (Tukey's HSD test: *P < 0.05; ***P < 0.001; n.s., not significant). Note that these markers started to express around week 3 of sexualization, which was an obvious stage beyond the point of no return (Fig 1B). Representative whole-mount *in situ* hybridization patterns for the ventral and dorsal sides of worms are shown. The expression pattern was judged based on five and three replicates in the asexual and sexual worms, respectively. Signals were seen as blue/purple staining. The testis, a copulatory apparatus, and the genital pore are indicated by blue, green, and pink arrowheads, respectively. A scale bar, 1 mm.
(TIF)

**S8 Fig. Alignment of predicted amino acid sequences of innexins.** The multiple sequence alignment analysis was performed using the GENETYX-MAC Version 13.0.14 (GENETYX, Tokyo, Japan) according to Nogi and Levin (2005). Sequence alignment of innexin proteins showing highly conserved regions. Black bars indicate the predicted transmembrane domains TM1–TM4. Red arrowheads indicate conserved cysteine residues in the extracellular loops. The matching amino acids are indicated using black shading, and low consensus (≥50%) amino acids are indicated using grey shading. (A) First transmembrane domains and the N-terminal and C-terminal flanking regions. (B) Conserved regions in the first extracellular loops. (C) Second transmembrane domains and the conserved amino acids YYQW(V) at the end of the first extracellular loops. The conserved amino acids are indicated using a green bar. (D) Third transmembrane domains and the C-terminal flanking regions. (E) Conserved regions in the second extracellular loops. (F) A part of the fourth transmembrane domains and the conserved region in the second extracellular loops. *Dr-siri* (TR37455|c0_g1_i1) in this study is indicated in red, and other innexin genes in *D. japonica* and *D. ryukyuensis* are indicated in green and blue, respectively. Dm, *Drosophila melanogaster*; Ce, *Caenorhabditis elegans*; Dj, *Dugesia japonica*; *Dr, Dugesia ryukyuensis*. Accession numbers: Dm_org_inx1, NP_524824; Dm_inx2, NP_572375; Dm_inx3, NP_524730; Dm_zpg_inx4, NP_648049; Dm_inx5; NP_573353; Dm_inx6, NP_572374; Dm_inx7, NP_788872; Dm_shakB_inx8, NP_728361; Dm_passove, AAA28745; Ce_eat-5, NP_492068, Ce_unc-7, NP_001257255; Ce_unc-9, NP_741917; Ce_inx-1, NP_741826; Ce_inx-2, NP_509885; Ce_inx-3, NP_509002;

Ce_inx-5, NP_509403; Ce_inx-6, NP_502435; Ce_inx-7, NP_500894; Ce_inx-8, NP_502209; Ce_inx-9, NP_502210; Ce_inx-10, NP_001024139; Ce_inx-11, NP_001256426; Ce_inx-12, NP_491213; Ce_inx-13, NP_491212; Ce_inx-14, NP_492078; Ce_inx-15, NP_491313; Ce_inx-16, NP_491314; Ce_inx-17, NP_491315; Ce_inx-18, NP_741294; Ce_inx-19, NP_490983; Ce_inx-20, NP_001251236; Ce_inx-21, NP_491187; Ce_inx-22, NP_491186; Dj_inx2, BAE78811; Dj_inx3, BAE78812; Dj_inx4, BAE78813; Dj_inx5, BAE78814; Dj_inx7, BAE78815; Dj_inx8, BAE78816; Dj_inx9, BAE78817; Dj_inx10, BAE78818; Dj_inx11, BAE78819; Dj_inx12, BAD83778 and Dj_inx13, BAE78820. See S7 Table for the information on innexins in *Dugesia ryukyuensis*.
(TIF)

**S9 Fig.  Phylogenic tree for innexins.** The UPGMA tree was constructed using MEGA version 12.0.11 according to Nogi and Levin (2005). The percentage of trees in which associated taxa clustered together in the bootstrap (1000 replicates) is shown next to the branches. A scale bar: Jaccard similarity coefficient. The predicted amino acid sequences of the conserved region, including the whole of the 1st–4th transmembrane domains (S8A–F Fig), were used for this analysis. *Dr-siri* (TR37455|c0_g1_i1) in this study is indicated in red, and other innexin genes in *D. japonica* and *D. ryukyuensis* are indicated in green and blue, respectively. The three groups of the planarian innexin sequences classified by this analysis (Nogi and Levin, 2005) are indicated by the bars and the names of the groups. Note that Dr-siri (TR37455|c0_g1_i1) was classified into Group III. Dm, *Drosophila melanogaster*; Ce, *Caenorhabditis elegans*; Dj, *Dugesia japonica*; *Dr, Dugesia ryukyuensis*. Accession numbers: Dm_org_inx1, NP_524824; Dm_inx2, NP_572375; Dm_inx3, NP_524730; Dm_zpg_inx4, NP_648049; Dm_inx5; NP_573353; Dm_inx6, NP_572374; Dm_inx7, NP_788872; Dm_shakB_inx8, NP_728361; Dm_passove, AAA28745; Ce_eat-5, NP_492068, Ce_unc-7, NP_001257255; Ce_unc-9, NP_741917; Ce_inx-1, NP_741826; Ce_inx-2, NP_509885; Ce_inx-3, NP_509002; Ce_inx-5, NP_509403; Ce_inx-6, NP_502435; Ce_inx-7, NP_500894; Ce_inx-8, NP_502209; Ce_inx-9, NP_502210; Ce_inx-10, NP_001024139; Ce_inx-11, NP_001256426; Ce_inx-12, NP_491213; Ce_inx-13, NP_491212; Ce_inx-14, NP_492078; Ce_inx-15, NP_491313; Ce_inx-16, NP_491314; Ce_inx-17, NP_491315; Ce_inx-18, NP_741294; Ce_inx-19, NP_490983; Ce_inx-20, NP_001251236; Ce_inx-21, NP_491187; Ce_inx-22, NP_491186; Dj_inx2, BAE78811; Dj_inx3, BAE78812; Dj_inx4, BAE78813; Dj_inx5, BAE78814; Dj_inx7, BAE78815; Dj_inx8, BAE78816; Dj_inx9, BAE78817; Dj_inx10, BAE78818; Dj_inx11, BAE78819; Dj_inx12, BAD83778 and Dj_inx13, BAE78820. See S7 Table for the information on innexins in *Dugesia ryukyuensis*.
(TIF)

**S10 Fig.  *Dr-siri* is expressed in the mesenchymal cells around the testes.** This result of FISH analysis is the low magnification image in Fig 13A. The mesenchymal cells expressing *Dr-siri* (green) around the testes can be recognized (white arrowheads). (a) The nuclei were counterstained with Hoechst 33342 (blue). (a') *Dr-siri* (green), (a") *Dr-klf4l* (magenta), and (a"') the merged images are shown. Domains bound by the blue line are testes.
(TIF)

**S11 Fig.  Homologs of *D. ryukyuensis* for neuro peptide Y genes and neuro peptide Y receptor-1 gene in *S. mediterranea*.** (A) We performed a local BLASTX search for the transcriptome catalogues of *D. ryukyuensis* (Sekii et al., 2019) by using eleven *npy* genes and an *npyr* gene of *S. mediterranea*. Top hits obtained in the BLAST search are shown. (B) The FPKM values in asexual and sexual worms of the homologs. Bar graphs were plotted based on FPKM value in the RNA-seq data of *D. ryukyuensis* (Sekii et al., 2019). Two races of sexual worms of *D. ryukyuensis* occur (Kobayashi et al., 2012). Ac-Sexual: Acquired sexual worms can switch to an asexual state; In-Sexual: Innate sexual worms cannot switch to an asexual state. (C) Whole-mount *in situ* hybridization patterns for *Dr-npy-6* and *Dr-npy-7* are shown. The expression pattern was judged based on five and three replicates in the asexual and sexual worms, respectively. Signals were seen

as blue/purple staining. Yellow, blue, and green arrowheads indicate a brain, ventral nerve cord, and copulatory apparatus, respectively. A scale bar, 2 mm.
(TIF)

**S1 Table. Sample information for the RNA-seq library focusing on the point of no return.**
(XLSX)

**S2 Table. Summary-1 of RNA-seq and mapping statistics.**
(XLSX)

**S3 Table. Differentially expressed genes with coding DNA sequences between the control and week 2 before-worms.**
(XLSX)

**S4 Table. Sample information in RNAi knockdown experiment of *Dr-nhr-1*, *Dr-dmd-1*, and *Dr-klf4l*.**
(XLSX)

**S5 Table. Sample information for RT-qPCR analysis in the RNAi knockdown experiment.**
(XLSX)

**S6 Table. Summary-2 of RNA-seq and mapping statistics.**
(XLSX)

**S7 Table. Innexin homologs in *Dugesia ryukyuensis.***
(XLSX)

**S8 Table. Sample information in RNAi knockdown experiment of *Dr-siri.***
(XLSX)

**S9 Table. Sample information for RT-qPCR analysis in the RNAi knockdown experiment of *Dr-siri.***
(XLSX)

**S10 Table. Oligonucleotide primers used in this study.**
(XLSX)

**S1 File. Raw data and statistical results for RT-qPCR analysis related to Fig 3.**
(XLSX)

**S2 File. Raw data and statistical results for RT-qPCR analysis related to Fig 4.**
(XLSX)

**S3 File. Raw data and statistical results for RT-qPCR analysis related to Fig 6.**
(XLSX)

**S4 File. Raw data and statistical results for RT-qPCR analysis related to Fig 7.**
(XLSX)

**S5 File. Raw data and statistical results for RT-qPCR analysis related to Fig 10.**
(XLSX)

**S6 File. Raw data and statistical results for RT-qPCR analysis related to Fig 11.**
(XLSX)

## Acknowledgments

We thank Dr. Hidefumi Orii, Hirosaki University, for advice concerning FISH methods. This work was done in part at the Gene Research Center, Hirosaki University.

## Author contributions

**Conceptualization:** Kazuya Kobayashi.

**Formal analysis:** Katsushi Yamaguchi, Shuji Shigenobu, Ryohei Furukawa.

**Funding acquisition:** Kazuya Kobayashi.

**Investigation:** Nobuyoshi Kumagai, Michio Kuroda, Tosei Hanai, Masaki Fujita, Takaaki Hino, Shunta Yorimoto, Sayaka Manta, Shuzo Nakagawa, Moe Yokoyama, Leon Tajima, Riku Ito, Hikaru Yamada, Kota Miura.

**Project administration:** Kazuya Kobayashi.

**Resources:** Makoto Kashima, Katsushi Yamaguchi, Shuji Shigenobu.

**Supervision:** Shuji Shigenobu, Ryohei Furukawa, Kiyono Sekii.

**Writing – original draft:** Nobuyoshi Kumagai.

**Writing – review & editing:** Kiyono Sekii, Kazuya Kobayashi.

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
