## [Decision Letter · Decision Letter 0]

17 Jul 2025

PGENETICS-D-25-00534

Identification of an innexin required for termination of the asexual state in planarian reproductive switching

PLOS Genetics

Dear Dr. Kobayashi,

Thank you for submitting your manuscript to PLOS Genetics. After careful consideration, we are happy to say that your manuscript has merit but does not fully meet PLOS Genetics's publication criteria as it currently stands. Therefore, we invite you to submit a revised version of the manuscript that addresses the points raised during both reviewers.

In particular, we agree with reviewer 2, that your results do not demonstrate conclusively that masculinization is crucial for switching from asexual to sexual reproduction.

Please submit your revised manuscript within 30 days Aug 16 2025 11:59PM. If you will need more time than this to complete your revisions, please reply to this message or contact the journal office at plosgenetics@plos.org. Please include the following items when submitting your revised manuscript:

We look forward to receiving your revised manuscript.

Kind regards,

Jean-René Huynh

Academic Editor

PLOS Genetics

Paula Cohen

Section Editor

PLOS Genetics

Aimée Dudley

Editor-in-Chief

PLOS Genetics

Anne Goriely

Editor-in-Chief

PLOS Genetics

**Journal Requirements:**

At this stage, the following Authors/Authors require contributions: Nobuyoshi Kumagai, Michio Kuroda, Tosei Hanai, Masaki Fujita, Takaaki Hino, Shunta Yorimoto, Sayaka Manta, Shuzo Nakagawa, Moe Yokoyama, Leon Tajima, Riku Ito, Hikaru Yamada, Kota Miura, Makoto Kashima, Katsushi Yamaguchi, Shuji Shigenobu, Ryohei Furukawa, and Kiyono Sekii. Please ensure that the full contributions of each author are acknowledged in the "Add/Edit/Remove Authors" section of our submission form.

The list of CRediT author contributions may be found here: https://journals.plos.org/plosgenetics/s/authorship#loc-author-contributions

Potential Copyright Issues:

i) Please confirm (a) that you are the photographer of 7, 12A, and S7, or (b) provide written permission from the photographer to publish the photo(s) under our CC BY 4.0 license.

ii) Figures 1A, and S1. Please confirm whether you drew the images / clip-art within the figure panels by hand. If you did not draw the images, please provide (a) a link to the source of the images or icons and their license / terms of use; or (b) written permission from the copyright holder to publish the images or icons under our CC BY 4.0 license. Alternatively, you may replace the images with open source alternatives. See these open source resources you may use to replace images / clip-art:

5) We note that your Data Availability Statement is currently as follows: "All data generated are included in the figures and supplemental files. Sequencing reads are available at the NCBI.". Please confirm at this time whether or not your submission contains all raw data required to replicate the results of your study. Authors must share the “minimal data set” for their submission. PLOS defines the minimal data set to consist of the data required to replicate all study findings reported in the article, as well as related metadata and methods (https://journals.plos.org/plosone/s/data-availability#loc-minimal-data-set-definition).

2) If any authors received a salary from any of your funders, please state which authors and which funders..

**Reviewers' comments:**

Reviewer's Responses to Questions

**Comments to the Authors:**

Reviewer #1: This article uses an assay previously established by the lab to study the induction of sexualization in the flatworm Dugesia ryukyuensis, a planarian species capable of switching between asexual and sexual reproduction. The assay marks the “point of no return,” the stage at which planarians, after transitioning to a sexual state, can no longer revert to a sexual state. By meticulously tracking the planarians for signs of sexual maturation and carefully identifying this critical point both morphologically and using several molecular markers, the authors performed RNA-sequencing analysis to identify genes involved at the point of no return. They identified several genes with homologs in the well-established planarian model Schmidtea mediterranea, which play a role in this transition point. Additionally, the authors discovered a new gene encoding an innexin, a protein responsible for forming gap junctions. The authors speculate that this gene may play a key role in the transport of sex-inducing substances. This is a rigorous study that opens new avenues for investigating how planarians undergo reproductive switching.

I have only minor concerns that should be easily addressed:

Line 468: The authors state that Dr-nanos expression in Dr-dmd-1(RNAi) animals is similar to controls, but significantly reduced in both Dr-nhr-1(RNAi) and Dr-klf4l(RNAi) animals. They also show that Dr-klf4l does not differ significantly in Dr-dmd-1(RNAi) animals (Figure 8). However, if no testes are observable in Dr-dmd-1(RNAi) animals (line 440), why isn’t the expression of Dr-nanos and Dr-klf4l reduced in these animals, given that they both are expressed in testis germ cells?

The FISH images in Figure 13 are unclear and should be repeated. The figure does not convincingly show that Dr-siri is expressed in both Dr-kl4l, Dr-nanos, and Dr-dmd-1-expressing cells (i.e. in both germ cells and testis niche cells). At a minimum, the authors should show that their FISH method can clearly distinguish between the expression of Dr-nanos/klf4l and Dr-dmd-1, which should not appear to co-express, before drawing strong conclusions about where Dr-siri is expressed using this technique. Either way, it is not necessary for Dr-siri to be expressed in both germ cells and niche cells (for gap junction formation between these cells) for the model to be correct. Even if it is only expressed in one cell type in the testis, other innexins could be expressed in the other cell type to facilitate gap junctional communication.

Reviewer #2: Kumagai et al use RNAseq, combined with dissection and regeneration experiments, to identify three genes that are linked with testicular differentiation and potentially the transition from asexual to sexual state in the planarian Dugesia ryukyuensis. RNAi assays were then conducted wherein worms were fed RNAi food and "sex inducing substances", finding that Dr-dmd-1-i worms developed ovaries but not testes, Dr-nhr-1-i worms failed to develop either, and Dr-klf4l-i worms developed small ovaries and no testes. Cut regenerated worms that were previously fed dsDNA did not continue to develop reproductive organs and returned to asexual state. Next, using RNAseq of the RNAi worms, they identify 27 differentially expressed genes, including an innexin that they name Dr-siri. RNAi of Dr-siri reveals that ovary and vitallaria develop, but not testes, and ultimately regenerates derived from the RNAi return to asexual state. Overall I find that the authors do a nice job identifying these genes and describing their potential function in D. ryukyuensis, cleverly taking advantage of the regeneration capacity of this species. I have some major and minor comments, however, that are outlined below:

It is important to be sure of the identity of each of these genes when assigning names and determining homology. Neighbour Joining methods are quick but not sufficiently robust and reliable, and a maximum likelihood approach should be used in addition to the neighbour joining trees (the software IQ-Tree, for example, has good documentation and offers model finding). Additionally, it is important where possible to include validated genes from multiple species in addition to Schmidtea mediterranea, C. elegans etc. Please also include bootstraps values on your trees.

The authors conclude that masculinization is crucial for switching from asexual to sexual reproduction, however I think this is an overstatement and do not think the results conclusively show this as they do not test the alternative. For example, what happens if genes involved in ovary/oocyte differentiation are knocked down such that testes develop normally? The discussion should be reworded accodingly, with the addition of "might" for example.

S1 Fig. This figure helps a lot to clarify the approach. However, if I understood correctly, half the head and half the tail fragments are left to regenerate, and the respective other halves are used to extract RNA. Once the sexual state is determined, then the RNA extractions are labelled as either asexual or sexual and pooled with RNA samples of the same category. The figure only shows heads, which is a bit confusing and I think it could help to add tails to this figure.

Fig 6 the top row label was confusing at first and the figure legend did not help. To help guide the reader, you could say explicitly that the top row refects expression of the gene targetted by RNAi respectively.

line 366 "RNA treatment was stated" perhaps this is a typo for started. 'Initiated with' could be an appropriate substitute

Line 366 The experimental approach is not completely clear to me. The worms are cut into three pieces, what happens to the other pieces (non pharangeal)? Are these discarded? And line 376, just to be sure that you mean that half is left to regenerate and half are taken for RNA extraction .

line 459 "No phenotypes were observed in the knockdown other than Dr-nhr-1, Dr-dmd-1, Dr-klf4l, and TR29311|c0_g1_i1 (S7 Fig), RNAi of which died because of curling and lysis phenotypes, regardless of sexualization." I do not understand this sentence. Dr-nhr-1, Dr-dmd-1 and Dr-klf4l are genes, not phenotypes. I also cannot understand the statement that knockdown of these genes led to death, given the description of the phenotypes in the previous two paragraphs. I also have trouble understanding S7 Fig and its relation to this statement. The figure legend does not seem to match the figure. Is this the correct figure legend?

**Have all data underlying the figures and results presented in the manuscript been provided?**

Reviewer #1: Yes

Reviewer #2: **No: ** The link to the DNA Databank where the data is stored did not work and I think the accession number needs to be updated.

PLOS authors have the option to publish the peer review history of their article (what does this mean? ). If published, this will include your full peer review and any attached files.

**Do you want your identity to be public for this peer review?** For information about this choice, including consent withdrawal, please see our Privacy Policy .

Reviewer #1: No

Reviewer #2: No

**Figure resubmission:**
---

## [Decision Letter · Decision Letter 1]

30 Oct 2025

Dear Dr Kobayashi,

We are pleased to inform you that your manuscript entitled "Identification of an innexin required for termination of the asexual state in planarian reproductive switching" has been editorially accepted for publication in PLOS Genetics. Congratulations!

Yours sincerely,

Jean-René Huynh

Academic Editor

PLOS Genetics

Paula Cohen

Section Editor

PLOS Genetics

Aimée Dudley

Editor-in-Chief

PLOS Genetics

Anne Goriely

Editor-in-Chief

PLOS Genetics

BlueSky: @plos.bsky.social

Comments from the reviewers (if applicable):

Reviewer's Responses to Questions

**Comments to the Authors:**

Reviewer #1: I appreciate the authors’ revisions. They have addressed my previous comments and the manuscript is now clearer. I also appreciate the effort to address my concerns with the quality of their whole-mount fluorescent in situ hybridizations, particularly given how technically challenging whole-mount FISH seems to be in these animals. This is a thorough study that provides valuable insight into the timing and molecular basis of sexualization in the flatworm Dugesia ryukyuensis. I am satisfied with the revisions and recommend the manuscript for acceptance.

- Minor typos on lines 98, 132, 134, 137, 939, 1072: Change “aPCSs” to “aPSCs”

- Figure 7 doesn’t need to be a stand-alone figure and seems like it would fit nicely as part of Figure 5.

Reviewer #2: The authors have satisfactorily responded to both my comments and those of the other reviewer, including re-running FISH analyses and also providing robust bootstrap supported gene trees. The adjustment of the language from 'masculinization' to testes formation appropriately reflects the results. The figure compression artifacts in the previous submission are resolved. I do not see any additional experiments or changes to be made to the manuscript.

**Have all data underlying the figures and results presented in the manuscript been provided?**

Reviewer #1: Yes

Reviewer #2: Yes

PLOS authors have the option to publish the peer review history of their article (what does this mean? ). If published, this will include your full peer review and any attached files.

**Do you want your identity to be public for this peer review?** For information about this choice, including consent withdrawal, please see our Privacy Policy .

Reviewer #1: No

Reviewer #2: No

**Data Deposition**

http://datadryad.org/submit?journalID=pgenetics&manu=PGENETICS-D-25-00534R1

**Press Queries**

---

## [Editor Report · Acceptance letter]

PGENETICS-D-25-00534R1

Identification of an innexin required for termination of the asexual state in planarian reproductive switching

Dear Dr Kobayashi,

We are pleased to inform you that your manuscript entitled "Identification of an innexin required for termination of the asexual state in planarian reproductive switching" has been formally accepted for publication in PLOS Genetics! Your manuscript is now with our production department and you will be notified of the publication date in due course.

With kind regards,

Judit Kozma

PLOS Genetics

On behalf of:
